# SPARC: CONTINUAL LEARNING BEYOND EXPERIENCE REHEARSAL AND MODEL SURROGATES

## ABSTRACT

Continual learning (CL) has become increasingly important as deep neural networks (DNNs) are required to adapt to the continuous influx of data without retraining from scratch. However, a significant challenge in CL is catastrophic forgetting (CF), where learning new tasks erases previously acquired knowledge, either partially or completely. Existing solutions often rely on experience rehearsal or full model surrogates to mitigate CF. While effective, these approaches introduce substantial memory and computational overhead, limiting their scalability and applicability in real-world scenarios. To address this, we propose SPARC, a scalable CL approach that eliminates the need for experience rehearsal and full-model surrogates. By effectively combining task-specific working memories and task-agnostic semantic memory for cross-task knowledge consolidation, SPARC results in a remarkable parameter efficiency, using only 6% of the parameters required by full-model surrogates. Despite its lightweight design, SPARC achieves superior performance on Seq-TinyImageNet and matches rehearsal-based methods on various CL benchmarks. Additionally, weight re-normalization in the classification layer mitigates task-specific biases, establishing SPARC as a practical and scalable solution for CL under stringent efficiency constraints. [1]

## 1 INTRODUCTION

Deep neural networks (DNNs), driven by large datasets and sophisticated algorithms, have shown exceptional performance across numerous tasks, including speech translation (Barrault et al., 2023), sentiment analysis (Devlin et al., 2018), and object recognition (Kirillov et al., 2023). However, as the scale of data increases, it becomes crucial for these models to learn continuously rather than retraining from scratch. Traditional training approaches are tailored to static data distributions, limiting their ability to handle dynamic data. Continual learning (CL) (Parisi et al., 2019; Hadsell et al., 2020; Wang et al., 2023) addresses this by enabling models to incrementally acquire new knowledge over time. However, a significant challenge in CL is catastrophic forgetting (McClelland et al., 1995; McCloskey & Cohen, 1989), where learning new information leads to the deterioration of previously acquired knowledge. This issue is not unique to CL but also arises in multitask learning (Kudugunta et al., 2019) and supervised learning under domain shifts (Ovadia et al., 2019). As a result, catastrophic forgetting has emerged as a critical barrier to the effective deployment of DNNs in dynamic environments.

To mitigate catastrophic forgetting, several strategies such as experience rehearsal, weight regularization, and parameter isolation have been proposed. These methods aim to preserve previously learned knowledge while enabling the acquisition of new information. Experience rehearsal methods (Arani et al., 2022; Pham et al., 2021a; Bhat et al., 2023) utilize memory buffers and model surrogates to replay past experiences during training, mitigating forgetting. However, these approaches are impractical for memory-constrained environments, such as edge devices, where buffer size is limited. Similarly, weight regularization approaches (Zenke et al., 2017; Chaudhry et al., 2018; Li & Hoiem, 2017) rely on model surrogates in the form of frozen networks to consolidate past knowledge but often struggle in class-incremental learning (Class-IL) settings, where distinguishing between classes learned across tasks is challenging. Parameter isolation methods (Aljundi et al., 2017; Rusu et al., 2016) allocate distinct parameters for each task to prevent interference but require task identity during

---

[1]The code will be publicly open upon acceptance.

inference and can suffer from capacity saturation in long task sequences. The reliance on memory buffers and model surrogates in these approaches complicates scaling for real-world applications, where memory constraints are critical.

Biological systems, particularly the human brain, provide a compelling blueprint for continual learning without catastrophic forgetting. The brain demonstrates the ability to learn, adapt, and accumulate knowledge over time, even in the face of dynamic external changes (Hadsell et al., 2020; Kudithipudi et al., 2022). According to the complementary learning systems (CLS) theory (McClelland et al., 1995), the slow-learning neocortex and fast-learning hippocampus work together to facilitate complex behavior, allowing continual learning without explicit experience rehearsal. Inspired by this, several artificial systems have attempted to mimic the interaction between the neocortex and hippocampus by employing model surrogates (Arani et al., 2022; Cha et al., 2021). While effective, these approaches introduce significant memory and computational overhead, making them unsuitable for deployment on memory-constrained devices. Therefore, a key challenge in designing CL systems is to replicate the success of biological systems without the need for memory-intensive experience rehearsal and model surrogates.

To this end, we propose *Simple PArameter isolation in a Restricted Capacity (SPARC)*, a continual learning approach that eliminates the need for both experience rehearsal and full model surrogates. SPARC leverages parameter-efficient depth-wise separable convolutions to serve as task-specific working memories, capturing task-relevant information, while point-wise convolutions act as task-agnostic semantic memory, consolidating knowledge across tasks. Additionally, SPARC incorporates weight re-normalization in the classification layer to counteract task-specific biases, a common issue in parameter isolation methods where the model disproportionately favors more recent tasks. The overall architecture of SPARC, as shown in Figure 1, is simple in design and grows linearly with the number of tasks, maintaining scalability even in memory-constrained environments. By using only 6% of the parameters required by full-model surrogates (Arani et al., 2022), SPARC achieves superior performance on Seq-TinyImageNet. In summary, our contributions are:

- We introduce SPARC, a rehearsal-free parameter isolation approach designed for vision-based continual learning, without the need for full model surrogates. Through extensive experiments, we demonstrate that SPARC achieves competitive performance with rehearsal-based methods across several CL benchmarks.
- As part of parameter isolation, we augment SPARC with task-specific working memories (Section 3.1) and task-agnostic semantic memory (Section 3.2) to effectively consolidate information across tasks.
- We identify task-specific biases as a key challenge in parameter isolation methods and propose a simple weight re-normalization technique (Section 3.3) to mitigate this issue, improving performance in continual learning settings.

## 2 MODEL SURROGATE BOTTLENECK

In a general continual learning (CL) setup, a model $\Phi_\theta$ with parameters $\theta \in \mathbb{R}^{|\theta|}$ is required to sequentially learn $k$ tasks. A core challenge in CL arises from the inaccessibility of previous tasks' data during the learning of new tasks. This results in the well-known stability-plasticity trade-off: balancing the retention of consolidated knowledge (stability) with the flexibility to acquire new information (plasticity). Greater stability risks static knowledge, while increased plasticity can lead to unlearning, also known as catastrophic forgetting. One straightforward method to alleviate this trade-off is experience rehearsal (ER) (Ratcliff, 1990), where a memory buffer stores and replays samples from prior tasks alongside new ones. Empirical evidence (e.g., DER++ (Buzzega et al., 2020)) demonstrates that larger buffers reduce forgetting. However, maintaining a buffer can, in some cases, raise privacy concerns and increase resource overhead. In memory-constrained environments, smaller buffers can result in overfitting to the stored samples (Bhat et al., 2022). Similarly, generative replay methods (e.g., DRI (Wang et al., 2022b)) face challenges related to the accuracy of the generative models, including their own susceptibility to forgetting and limitations in expressiveness (Wang et al., 2023). To bypass the limitations of experience rehearsal, many approaches employ model surrogates. These methods seek to stabilize learning by maintaining auxiliary models, allowing the main model to focus on learning new tasks. Inspired by the Complementary Learning Systems (CLS) theory, works like CLS-ER (Arani et al., 2022), OCDNet (Li et al., 2022), and TAMiL (Bhat et al., 2023) use

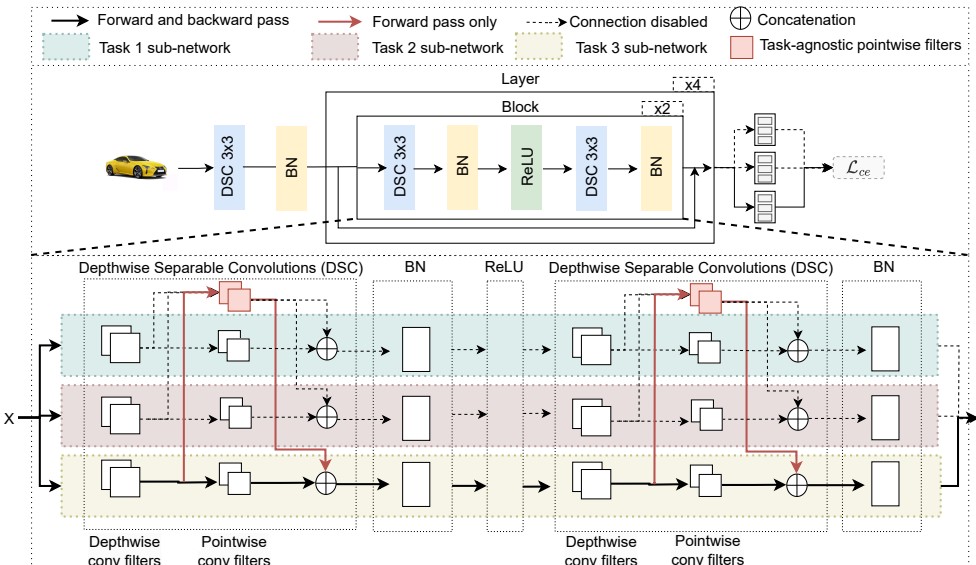

Figure 1: The SPARC architecture, using ResNet-18 of 4 layers with 2 blocks each. Task-specific working memories (shown in white) efficiently capture task-relevant information, while the task-agnostic semantic memory (highlighted in red) consolidates knowledge across tasks. This design enables SPARC to effectively balance plasticity and stability, achieving scalable continual learning without the need for full model surrogates or experience rehearsal.

an exponential moving average (EMA) of model weights as a slow-learning surrogate to consolidate task knowledge. While these approaches improve retention of past tasks, they introduce significant computational overhead due to the use of multiple model surrogates, making them less efficient for large-scale applications.

An alternative to surrogate-based methods is to frame CL within a Bayesian context. Given the current task data $\mathcal{D}_t$ and the prior $p\left(\theta \mid \mathcal{D}_{1:t-1}\right)$, the posterior distribution $p\left(\theta \mid \mathcal{D}_{1:t}\right)$ can be updated using Bayes' rule. Since computing the posterior directly is intractable, approximations like the online Laplace approximation or Fisher information matrix are employed (Ritter et al., 2018; Kirkpatrick et al., 2017). These methods effectively regularize weight updates by penalizing deviations from previous tasks' learned parameters. However, regularization-based methods still struggle with Class-IL because they fail to discriminate between classes across different tasks.

The use of repeated learning in a fixed-capacity model often leads to inter-task interference, where parameters allocated for one task interfere with those of others, reducing overall performance (Wang et al., 2023). Parameter isolation approaches aim to mitigate this interference by dedicating task-specific parameters $e^{(t)}$ while sharing task-agnostic parameters ($\psi$) across tasks. Methods like PNN (Rusu et al., 2016), DEN (Yoon et al., 2018), and CPG (Hung et al., 2019a) reduce inter-task interference by splitting model parameters, but they require task identity during inference and encounter scalability issues as the number of tasks grows. To address the limitations of task-specific parameter isolation, sparse dynamic parameter isolation methods have been proposed, such as PackNet (Mallya & Lazebnik, 2018) and NISPA (Gurbuz & Dovrolis, 2022). These methods draw inspiration from the brain's sparse connectivity, creating stable, task-specific paths within a fixed model capacity. By using sparse subsets of parameters, these models aim to preserve prior knowledge while acquiring new information. However, these methods also depend on task identity during inference and suffer from capacity saturation in long task sequences, which limits their scalability. Furthermore, they often require multiple model surrogates to manage task-specific parameter subsets, further complicating deployment in real-world applications. A comprehensive review of the related works can be found in the Appendix B.

In summary, model surrogates in CL come in various forms: be it (a) additional EMA models in rehearsal-based approaches, (b) a copy of the previous task model in weight-regularization approaches, or (c) large sub-network for each task in PNNs or even as many task masks as the number of tasks

in sparse dynamic parameter isolation approaches. The reductions in catastrophic forgetting in these approaches can often be correlated with either an increase in overall model size due to model surrogates and/or memory buffer size. Most approaches discussed in this paper utilize full model surrogates and/or experience rehearsal to some extent. To this end, we attempt to highlight the problem of correlation between model size and/or increase in buffer size with the reduction in catastrophic forgetting in CL. In light of these challenges, we propose SPARC, a parameter-efficient, scalable, and rehearsal-free parameter isolation method that addresses Class-IL without relying on full model surrogates or explicit experience rehearsal. SPARC consolidates knowledge across tasks while avoiding the computational and memory bottlenecks inherent to existing surrogate-based approaches, making it well-suited for scalable CL in memory-constrained environments.

## 3 METHOD

A typical CL setup consists of $k$ sequential tasks where the model is expected to learn a new task $t$ while retaining information from previous tasks. CL is particularly challenging for SPARC, as access to the previous data distributions $\{\mathcal{D}_1, \ldots, \mathcal{D}_{t-1}\}$ is completely restricted when learning a new task. In other words, SPARC does not rely on experience rehearsal. As a result, optimizing the CL model $\Phi_\theta$ using only the cross-entropy objective for the current task can excessively favor plasticity over stability, leading to overfitting on the current task and catastrophic forgetting of prior tasks. To address this, our CL model $\Phi_\theta$, parameterized as $\theta = \cup_{t=1}^{k} \theta^{(t)} = \cup_{t=1}^{k} \{f_{\theta^t}, g_{\theta^t}, \psi_\theta\}$, consists of a disjoint set of task-specific parameters (working memories). For each task $t$, the feature extractor $f_{\theta^t}$ and classifier $g_{\theta^t}$ are learned through task-specific parameters $\theta^{(t)}$, while task-agnostic parameters $\psi_\theta$ (semantic memory) facilitate knowledge consolidation across tasks. In the following subsections, we describe how parameter isolation is enforced within different layers and explain the mechanisms that enable effective information consolidation and weight re-normalization.

### 3.1 TASK-SPECIFIC LEARNING THROUGH WORKING MEMORIES

We assume prior knowledge of the task boundary information to allocate a new sub-network for each task. Most CL approaches utilize ResNet-18 (He et al., 2016) as their backbone for empirical studies. At its core, ResNet-18 features convolutional layers organized into 4 residual blocks, using skip connections to facilitate the training of deep networks. These blocks also include batch normalization (BN) and rectified linear unit (ReLU) activation functions. Additionally, ResNet-18 incorporates pooling layers for downsampling feature maps, and a fully connected layer for final classification. However, repeated learning within a fixed-capacity network leads to significant inter-task interference (Wang et al., 2023). Moreover, parameter isolation using traditional convolutional layers becomes unscalable in long task sequences, as seen in models such as PNNs. The non-stationary nature of CL data further exacerbates the mismatch between training and testing in BN layers (Pham et al., 2021b).

In SPARC, we address these challenges by utilizing task-specific working memories for each task. Apart from the task-agnostic parameters, each working memory is self-contained with its own convolutional, BN, and classification layers. Within each working memory, we replace traditional convolutional layers with parameter-efficient and computationally cheaper depth-wise separable convolutions (DSCs) (Chollet, 2017; Howard et al., 2018; Guo et al., 2019). DSCs consist of two operations: depth-wise convolution, which applies spatial convolution independently to each input channel, and point-wise convolution, which projects the depth-wise output onto a new channel space. These operations are described as:

$$\hat{O}_{h,l,m} = \sum_{i,j} \hat{K}_{i,j,m}^t \cdot F_{h+i-1,l+j-1,m} \tag{1}$$

$$O_{h,l,n} = \sum_{m} \tilde{K}_{m,n}^t \cdot \hat{O}_{h-1,l-1,m} \tag{2}$$

where $F$ and $O$ represent the input and output feature maps, respectively, and $\hat{K}$ and $\tilde{K}$ denote the depth-wise and point-wise filters. The indices $h$, $l$, and $m$ correspond to the spatial height, spatial width, and channel of the feature map, respectively. Similarly, $i$, $j$, and $n$ represent the offsets in the height and width dimensions and the output channel index, respectively. For each task, a set of depth-wise and point-wise filters is isolated and updated independently from other tasks' parameters.

The choice of DSC over traditional convolutions serves several purposes: (i) Efficiency: DSCs capture the most significant components of traditional convolutions while discarding redundant information, making them computationally efficient (Guo et al., 2018). (ii) Parameter Efficiency: By reducing over-parameterization, DSCs introduce implicit regularization, which helps prevent overfitting and improves generalization. (iii) Scalability: Parameter isolation using DSCs remains scalable even as the number of tasks increases. More details on DSCs are provided in Appendix C.

To mitigate the training and testing discrepancy in BN layers, SPARC maintains task-specific $\gamma$ and $\beta$ parameters (the learnable vectors in BN) along with running estimates of the mean and variance for each working memory. This segregated normalization facilitates parameter isolation during training while ensuring proper normalization during inference (Pham et al., 2021b) by applying task-specific moments to task-specific input features.

Finally, in the fully connected (FC) classification layer, SPARC allocates a subset of neurons for each task based on the number of classes, along with their corresponding incoming connections. Connections between neurons belonging to different tasks, referred to as cross-task connections, are discarded to avoid interference, preserving the stability of the CL model. Essentially, each task operates with an isolated fully connected layer serving as its classification layer.

## 3.2 TASK-AGNOSTIC SEMANTIC INFORMATION CONSOLIDATION

Hard parameter isolation within each working memory has a downside: the number of parameters increases significantly as the number of tasks grows. Conversely, maintaining model compactness through parameter sharing is not ideal either, as repeated learning on shared parameters results in higher forgetting. To strike a better balance between model compactness and performance, we introduce a task-agnostic shared semantic memory that consolidates knowledge across tasks while minimizing forgetting.

We achieve parameter-efficient task-agnostic information consolidation by sharing a portion of the point-wise filters across tasks. Specifically, half of the point-wise filters remain task-specific, while the other half are shared among tasks. This modifies the point-wise operation in Eqn. 2 as follows:

$$O_{h,l,n} = \begin{cases} \sum_m \tilde{K}^t_{m,n} \cdot \hat{O}_{h-1,l-1,m}, & \text{if } n \leq N/2, \\ \sum_m \tilde{K}^c_{m,n} \cdot \hat{O}_{h-1,l-1,m}, & \text{if } n > N/2. \end{cases} \tag{3}$$

where $\tilde{K}^t$ and $\tilde{K}^c$ denote the task-specific and task-agnostic point-wise filters, respectively, and $N$ represents the total number of point-wise filters. Note that the outputs of the task-specific and task-agnostic filters are concatenated. The task-agnostic filters $\tilde{K}^c \in \psi_\theta$ are randomly initialized and learned during the first task, then updated as an exponential moving average of the previous task's filters $\tilde{K}^{t-1}$ at the end of each task, defined as:

$$\tilde{K}^c = \alpha \, \tilde{K}^c + (1 - \alpha) \, \tilde{K}^{t-1} \quad \forall \, t > 2 \tag{4}$$

As $\tilde{K}^c$ consolidates information across tasks, it enhances the current task's performance without compromising the stability of previous tasks. This task-agnostic information consolidation enables SPARC to remain parameter-efficient while closely approximating the performance of hard parameter isolation (see Table 5). Similar to the CLS theory, where working and semantic memories complement each other, SPARC's working memories capture task-specific information, while its semantic memory captures task-agnostic information, achieving an effective balance between plasticity and stability.

## 3.3 WEIGHT RE-NORMALIZATION

In CL, the sequential nature of task learning often results in higher weight magnitudes for the classification layer of later tasks, leading to stronger activations and task recency bias in Class-IL (Zhao et al., 2020). In parameter isolation methods like SPARC, the isolated training approach can amplify task-specific biases, causing decisions to favor certain tasks while reducing clarity for others. To address the weight magnitude disparity in the classification layer, we propose a weight re-normalization technique based on activation-derived normalization constants.

Let $A^t = \{max(g_{\theta^t}(.))\}$ denote the set of maximum activations from the fully connected (FC) layer of the current task over all samples during the final training epoch. Define $A^t_{0.25} = \text{Quartile}(A^t, 0.25)$

and $A_{0.75}^t = \text{Quartile}(A^t, 0.75)$ as the first and third quartiles of this set, and let $A_{IQR}^t = A_{0.75}^t - A_{0.25}^t$ be the inter-quartile range (IQR). At the end of training for each task, the task-specific weights and biases of the FC layer are re-normalized as follows:

$$W_{\hat{t}} = \frac{\kappa \cdot W_t}{\eta}, \quad B_{\hat{t}} = \frac{\kappa \cdot B_t}{\eta}, \quad \eta = \max\{a \in A^t \mid a \le (A_{0.75}^t + A_{IQR}^t)\} \tag{5}$$

Here, $\kappa$ is a constant, set to 5 in our experiments. This weight re-normalization method is straightforward and does not require any additional validation sets or model parameters, effectively reducing weight magnitude disparity in the classification layer and mitigating task-specific biases in SPARC.

### 3.4 PUTTING IT ALL TOGETHER

SPARC is a simple, rehearsal-free parameter isolation approach designed to address catastrophic forgetting in continual learning (CL). To effectively evaluate SPARC against comparable methods, we build its backbone similar to ResNet-18. As illustrated in Figure 1, SPARC consists of four layers, each containing two blocks, with standard convolutions replaced by depth-wise separable convolutions (DSCs). For the point-wise filters, half are task-specific, while the remaining are shared across tasks. Both task-specific and task-agnostic point-wise filters process the output from the depth-wise filters independently, and their outputs are concatenated (as shown in Eqn. 1 and 3). As CL exacerbates the mismatch between training and testing in BN layers, SPARC maintains task-specific BN layers along with their own running estimates of the mean and variance for each working memory. During training, task-specific data is accessed, and the corresponding sub-network including their respective BN layers are updated through gradient updates. The learning objective for each task is defined as:

$$\mathcal{L}_t = \mathbb{E}_{(x_i, y_i) \sim \mathcal{D}_t} \mathcal{L}_{ce}(\sigma(\Phi_{\theta^t}(x_i)), y_i), \tag{6}$$

where $\mathcal{L}_{ce}$ represents the cross-entropy loss, and $\Phi\theta^t$ is the model for task $t$. From the second task onward, task-agnostic shared parameters are updated using an exponential moving average (EMA) as described in Eqn. 4. We also monitor the highest activations in the classification layer during the final training epoch. After training each task, the task-specific weights and biases are re-normalized using Eqn. 5 to address weight magnitude disparity across tasks. For inference in the Class-IL setting, each image is independently processed through all sub-networks, including their respective batch normalization layers. The outputs of all sub-networks are then concatenated, and the class with the highest activation is selected (refer to Appendix A.1). This approach enables task-agnostic inference across multiple tasks. In the Task-IL setting, inference is restricted to the specific sub-network associated with the task, ensuring that only the task-relevant parameters are utilized.

## 4 RESULTS

**Experimental setup.** We evaluate SPARC in the contexts of Class-IL and Task-IL on Split-CIFAR10, Split-CIFAR100, Split-TinyImageNet, Split-MiniImageNet, and Seq-ImageNet100, averaging results over three runs. Several baselines are considered, representing different CL approaches: experience rehearsal, weight regularization, parameter isolation with fixed capacity, and growing architectures. More details on these baselines can be found in Appendix B. For comparison, we also include a lower-bound baseline, *SGD*, which lacks mechanisms to counteract catastrophic forgetting, and an upper-bound baseline, *Joint*, which is trained on the entire dataset simultaneously. While most baselines use ResNet-18 (He et al., 2016) as the backbone, SPARC utilizes a ResNet-18-like architecture with DSC layers. For each task, we reserve 32, 64, 128, and 256 depth-wise filters in layers 1 to 4, respectively. Further details on datasets, settings, evaluation metrics, and backbones can be found in Appendix F.1.

### 4.1 EMPIRICAL EVALUATION

Table 1 compares SPARC with various CL approaches. Weight regularization methods (e.g., LwF, SI, oEWC) perform moderately in Task-IL and poorly in Class-IL, even with a full model surrogate, as they prioritize stability over plasticity by copying previous task models. In contrast, SPARC ensures maximal stability through parameter-efficient, task-specific working memories without relying on full model surrogates or weight regularization.

Table 1: A benchmark comparison with prior works on Class-IL and Task-IL. The best results are in bold, and the second-best are underlined. The methods are divided into JOINT (upper bound) and SGD (lower bound), weight regularization, parameter isolation, and rehearsal-based with 200 buffer size. $\mathcal{F}$ and $\mathcal{B}$ indicate the number of forward and backward passes through the CL model, and #Params (M) indicates the number of parameters (in millions) used for Seq-CIFAR100 with 5 tasks. Refer to Appendix Section B for more details on competing methods (see Table 9 for references to methods.), Section F.2 for exceptions, Section D.1 for results with a buffer size of 500, and Section D.3 for evaluation on ImageNet subsets.

| Method | #Params (M) | # of $\mathcal{F}$ and $\mathcal{B}$ | Seq-CIFAR10 (5T) Class-IL | Task-IL | Seq-CIFAR100 (5T) Class-IL | Task-IL | Seq-TinyImageNet (10T) Class-IL | Task-IL |
|---|---|---|---|---|---|---|---|---|
| JOINT | 11.23 | 1$\mathcal{F}$, 1$\mathcal{B}$ | 92.20 ±0.15 | 98.31 ±0.12 | 70.56 ±0.28 | 86.19 ±0.43 | 59.99 ±0.19 | 82.04 ±0.10 |
| SGD | 11.23 | 1$\mathcal{F}$, 1$\mathcal{B}$ | 19.62 ±0.05 | 61.02 ±3.33 | 17.49 ±0.28 | 40.46 ±0.99 | 7.92 ±0.26 | 18.31 ±0.68 |
| oEWC | 22.46 | 2$\mathcal{F}$, 1$\mathcal{B}$ | 19.49 ±0.12 | 68.29 ±3.92 | - | - | 7.58 ±0.10 | 19.20 ±0.31 |
| SI | 22.46 | 2$\mathcal{F}$, 1$\mathcal{B}$ | 19.48 ±0.17 | 68.05 ±5.91 | - | - | 6.58 ±0.31 | 36.32 ±0.13 |
| ALASSO | 11.23 | 1$\mathcal{F}$, 1$\mathcal{B}$ | 25.19 | 73.79 | - | - | 17.02 | 48.07 |
| UCB | 11.23 | 1$\mathcal{F}$, 1$\mathcal{B}$ | 56.23 | 78.56 | - | - | 23.43 | 49.01 |
| BMKP | 11.23 | 3$\mathcal{F}$, 2$\mathcal{B}$ | - | 94.49 ±0.26 | - | - | - | 70.36 ±0.32 |
| PNNs | 216.7 | 1$\mathcal{F}$, 1$\mathcal{B}$ | - | 95.13 ±0.72 | - | 74.01 ±1.11 | - | **67.84** ±0.29 |
| PackNet | 33.6 | 1$\mathcal{F}$, 1$\mathcal{B}$ | - | 93.73 ±0.55 | - | 72.39 ±0.37 | - | 60.46 ±1.22 |
| NISPA | 8.75 | 1$\mathcal{F}$, 1$\mathcal{B}$ | - | 57.36 ±1.92 | - | 65.36 ±2.19 | - | 59.56 ±0.32 |
| SparCL-EWC | - | 1$\mathcal{F}$, 1$\mathcal{B}$ | - | 68.33 ±0.54 | - | 59.53 ±0.25 | - | 59.56 ±0.32 |
| ER | 11.23 | 1$\mathcal{F}$, 1$\mathcal{B}$ | 44.79 ±1.86 | 91.19 ±0.94 | 21.40 ±0.22 | 61.36 ±0.35 | 8.57 ±0.04 | 38.17 ±2.00 |
| DER++ | 11.23 | 2$\mathcal{F}$, 1$\mathcal{B}$ | 64.88 ±1.17 | 91.92 ±0.60 | 29.60 ±1.14 | 62.49 ±1.02 | 10.96 ±1.17 | 40.87 ±1.16 |
| ER-ACE | 11.23 | 1$\mathcal{F}$, 1$\mathcal{B}$ | 62.08 ±1.44 | 92.20 ±0.57 | 35.17 ±1.17 | 63.09 ±1.23 | 11.25 ±0.54 | 44.17 ±1.02 |
| Co$^2$L | 22.67 | 4$\mathcal{F}$, 1$\mathcal{B}$ | 65.57 ±1.37 | 93.43 ±0.78 | 31.90 ±0.38 | 55.02 ±0.36 | 13.88 ±0.40 | 42.37 ±0.74 |
| GCR | 11.23 | 1$\mathcal{F}$, 1$\mathcal{B}$ | 64.84 ±1.63 | 90.80 ±1.05 | 33.69 ±1.40 | 64.24 ±0.83 | 13.05 ±0.91 | 42.11 ±1.01 |
| CLS-ER | 33.69 | 3$\mathcal{F}$, 1$\mathcal{B}$ | 66.19 ±0.75 | 93.90 ±0.60 | 43.80 ±1.89 | 73.49 ±1.04 | 23.47 ±0.80 | 49.60 ±0.72 |
| OCDNet | 22.46 | 2$\mathcal{F}$, 1$\mathcal{B}$ | **73.38** ±0.32 | 95.43 ±0.30 | 44.29 ±0.49 | 73.53 ±0.24 | 17.60 ±0.97 | 56.19 ±1.31 |
| TAMiL | 23.10 | 2$\mathcal{F}$, 1$\mathcal{B}$ | 68.84 ±1.18 | 94.28 ±0.31 | 41.43 ±0.75 | 71.39 ±0.17 | 20.46 ±0.40 | 55.44 ±0.52 |
| TriRE | 100.98 | 2$\mathcal{F}$, 1$\mathcal{B}$ | 68.17 ±0.33 | 92.45 ±0.18 | 43.91 ±0.18 | 71.66 ±0.44 | 20.14 ±0.19 | 55.95 ±0.78 |
| **SPARC** | **1.04** | 1$\mathcal{F}$, 1$\mathcal{B}$ | 61.22 ±4.81 | **95.76** ±0.21 | **49.03** ±0.05 | **75.52** ±0.11 | **32.29** ±0.01 | 65.66 ±0.01 |

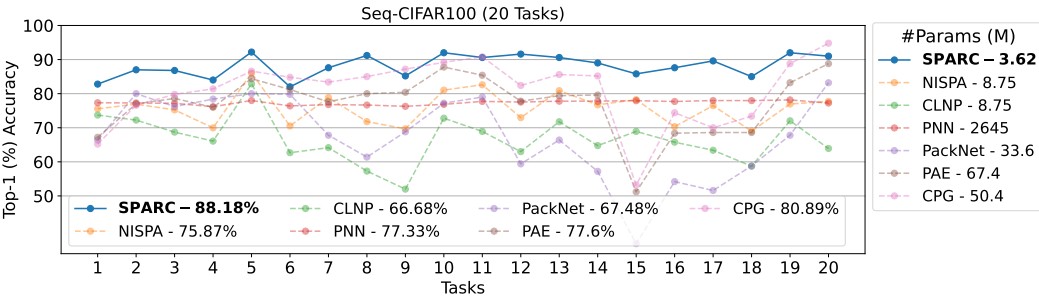

Figure 2: Comparison with parameter isolation approaches on Seq-CIFAR100 with 20 tasks. We report the final accuracy of each task after training on all tasks.

The performance of parameter isolation methods (e.g., PNNs, PackNet) in Task-IL is comparable to SPARC due to their over-parameterization. We also compare SPARC with rehearsal-based approaches using buffer sizes of 200 and 500 (Tables 1 and 10), including those relying solely on experience rehearsal (e.g., ER, DER++), approaches utilizing multiple model surrogates (e.g., CLS-ER, Co$^2$L, OCDNet, TAMiL), and methods incorporating generative replay (e.g., DRI). SPARC, unlike these methods, does not use experience rehearsal or full model surrogates to counter catastrophic forgetting. In simpler scenarios like Seq-CIFAR10, SPARC's performance is competitive but lags behind most rehearsal-based approaches. However, as the buffer-to-class ratio decreases and dataset complexity increases, the performance of rehearsal-based methods declines due to class under-representation in the buffer. With few exceptions, SPARC outperforms most competing methods in Seq-CIFAR100 and Seq-TinyImageNet scenarios with a buffer size of 200 and remains highly competitive with a buffer size of 500 (refer to Table 10).

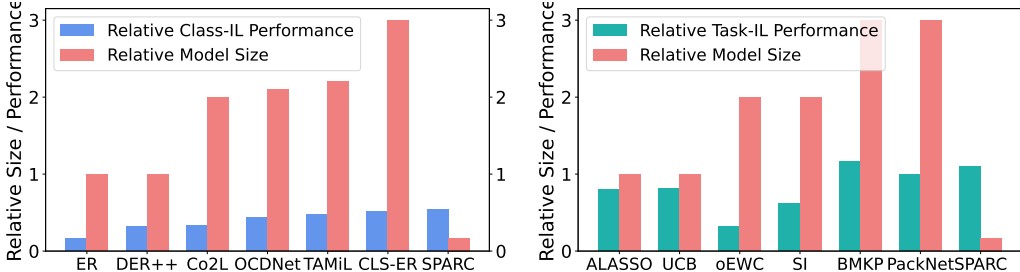

Figure 3: Comparison of relative performance and model size of different CL approaches in Seq-TinyImageNet 10 tasks with respect to a JOINT model in Class-IL (left) and Task-IL (right) settings.

Table 2: Effect of width and depth on SPARC in Seq-CIFAR100 with 5 Tasks.

| Width factor | Depth | #Filters per task | #Params (M) | Accuracy |
|---|---|---|---|---|
| 1/4 | 1 | [16] | 0.009 | 17.53 $\pm0.24$ |
| 1/4 | 2 | [16, 32] | 0.028 | 29.31 $\pm0.07$ |
| 1/4 | 3 | [16, 32, 64] | 0.087 | 41.19 $\pm0.02$ |
| 1/4 | 4 | [16, 32, 64, 128] | 0.291 | 45.24 $\pm0.19$ |
| 1/2 | 2 | [32, 64] | 0.083 | 37.81 $\pm0.47$ |
| 1/2 | 3 | [32, 64, 128] | 0.287 | 47.00 $\pm1.78$ |
| 1/2 | 4 | [32, 64, 128, 256] | 1.040 | 49.03 $\pm0.05$ |
| 1/4 | 4 | [16, 32, 64, 128] | 0.291 | 45.24 $\pm0.19$ |
| 1/2 | 4 | [32, 64, 128, 256] | 1.040 | 49.03 $\pm0.05$ |
| 1 | 4 | [64, 128, 256, 512] | 3.910 | 52.48 $\pm0.86$ |

Table 3: Performance evaluation on Seq-ImageNet100 with 10 tasks.

| Method | Incremental Accuracy (%) |
|---|---|
| LwF | 31.2 |
| EWC | 20.4 |
| MUC | 35.1 |
| LUCIR | 41.4 |
| SPARC | **50.90** |

Figure 2 compares SPARC with parameter isolation approaches (PNNs, CPG, PAE) and dynamic sparse methods (CLIP, NISPA, PackNet) on Seq-CIFAR100 across 20 tasks. The figure shows the final Task-IL accuracies after training on all tasks. While parameter isolation approaches grow beyond model capacity, dynamic sparse architectures learn task-specific masks within a fixed model capacity. SPARC strikes a balance between these methods, growing beyond model capacity, but more moderately than other parameter isolation approaches. As shown, SPARC achieves superior performance across tasks with a modest model size.

We also report the performance of various approaches on Seq-ImageNet100, divided into 10 tasks, in Table 3. Seq-ImageNet100 is a subset of ImageNet-1k with 100 classes evenly distributed across 10 tasks. With only 1.9 million parameters and without any dataset-specific hyperparameter tuning, SPARC demonstrates superior performance on Seq-ImageNet100. As seen later in Table 2, performance of SPARC can be further enhanced by increasing the model's width, depth, or both.

**Model size vs performance:** Ideally, a CL model should achieve performance comparable to the JOINT model while maintaining a model size that is equal to or smaller. However, many CL approaches rely on experience rehearsal and model surrogates to counter catastrophic forgetting, which leads to an increase in both the number of parameters and computational complexity. As shown in Figure 3, the reduction in catastrophic forgetting on Seq-TinyImageNet is largely due to the use of model surrogates. In stark contrast, SPARC achieves superior performance across most CL benchmarks while using only a fraction of the parameters. This makes SPARC a compelling choice for real-world applications where memory and compute resources are limited.

**Effect of width and depth:** We present an ablation study on the impact of width and depth on SPARC's performance. By default, SPARC's backbone has the same number of filters for 2 tasks per block as a standard ResNet-18. As the number of tasks increases beyond 2, SPARC becomes wider than ResNet-18. Thanks to the use of DSC layers, SPARC remains parameter-efficient even with a larger number of filters per block. Table 2 shows the results of varying the width and depth for SPARC. Consistent with the findings in Mirzadeh et al. (2022), increasing SPARC's width improves performance. Depth also significantly influences performance, though after a certain point, the improvements do not correspond to the increased number of parameters.

Table 4: Growth in number of parameters (millions) for different number of task sequences in Seq-CIFAR100.

| Methods | 5 tasks | 10 tasks | 20 tasks |
|---------|---------|----------|----------|
| ER | 11.23 | 11.23 | 11.23 |
| DER++ | 11.23 | 11.23 | 11.23 |
| CLS-ER | 33.69 | 33.69 | 33.69 |
| TAMiL | 23.10 | 23.76 | 25.08 |
| PNNs | 216.7 | 735.28 | 2645.05 |
| SPARC | **1.04** | **1.90** | **3.62** |

Table 5: Evaluation of semantic information consolidation in SPARC on Seq-CIFAR100 5 tasks.

| Method | #Param (M) | Class-IL |
|--------|-----------|----------|
| Shared point-wise & depth-wise filters | 0.33 | 22.37 ±0.07 |
| Shared point-wise & separate depth-wise filters | 0.43 | 42.77 ±0.28 |
| Semantic information consolidation (Sec. 3.2) | 1.04 | 49.13 ±0.25 |
| Separate point-wise & depth-wise filters | 1.65 | 51.57 ±0.27 |

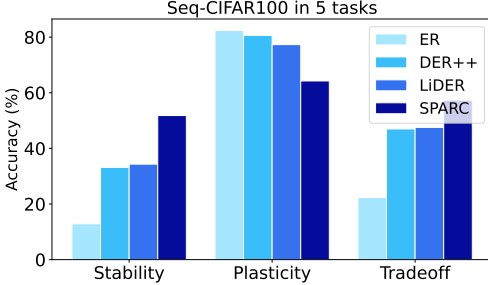
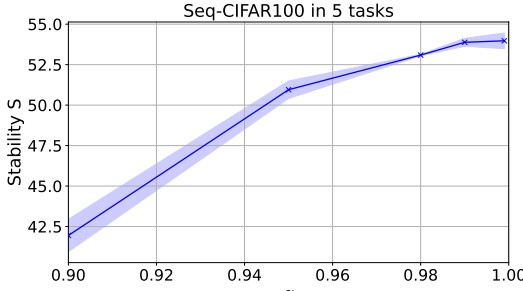

Figure 4: (Left) Stability-plasticity trade-off of different CL models on Seq-CIFAR100 5 tasks with buffer size 500. (Right) Effect of semantic information consolidation on SPARC's stability. $\mathcal{S}$ represents model stability at task $t$, quantified as the average performance across all preceding tasks.

**Parameter growth:** Table 4 compares parameter growth across various CL approaches with different task sequences. As shown, SPARC uses far fewer parameters compared to both rehearsal-based and parameter isolation methods. Moreover, SPARC remains scalable even with longer task sequences. In Seq-TinyImageNet with 10 tasks, SPARC outperforms CLS-ER using just 6% of the parameters without the need for experience rehearsal. Both CLS-ER and SPARC leverage complementary learning systems to consolidate knowledge across tasks, but CLS-ER relies on two full model surrogates, resulting in a large memory footprint. In contrast, SPARC creates an efficient combination of parameter-efficient working and semantic memories, all within a restricted model capacity.

**Stability-plasticity trade-off:** Paramount to CL is the stability-plasticity trade-off, a model's capacity to improve and acquire new knowledge and tasks while preserving performance on earlier learned abilities (Mermillod et al., 2013). This balance is critical for developing adaptable learning algorithms. Following (Sarfraz et al., 2022a), we evaluate stability-plasticity trade-off to better understand the ability of various methods to maintain this balance. Let $\mathcal{T}$ denote the task-wise performance matrix, where $\mathcal{T}_{i,j}$ signifies the accuracy on task $j$ after learning task $i$. The stability $\mathcal{S}$ of a model at task $t$ is quantified as the average performance across all preceding tasks, $mean(\mathcal{T}_{t,1:t,t-1})$. Conversely, the plasticity $\mathcal{P}$ at task $t$ is given by the average performance across tasks $1$ to $t$ when they are first learned, $mean(Diag(\mathcal{T}))$. The trade-off between stability and plasticity is subsequently defined as $Trade\text{-}off = 2\frac{SP}{S+P}$.

Figure 4 (left) presents an overview of stability-plasticity trade-off in Seq-CIFAR100 5 tasks with buffer size 500. As can be seen, SPARC is way more stable and moderately plastic due to parameter isolation and task-agnostic information consolidation. Such a setting allows SPARC to capture task-specific information with quite less number of parameters and retain them without catastrophic forgetting thereby managing this trade-off better.

**Effect of semantic information consolidation:** The task-agnostic semantic information consolidation presented in Section 3.2 positions SPARC to be as parameter-efficient as possible. Table 5 presents results on Seq-CIFAR100 5 tasks with three extremes: (i) All filters are shared across

tasks; (ii) Only point-wise filters are shared across tasks; and (iii) Each task entails separate filters. As can be seen, SPARC outperforms both shared versions by a large margin. Essentially, semantic memory helps consolidate information across tasks. On the other hand, SPARC almost matches the performance of separate point-wise filters while being parameter-efficient. Specifically, the separate point-wise filters version has **59%** more parameters with only **5%** relative improvement in performance. The difference in terms of the number of parameters will be even more pronounced in longer task sequences.

While semantic information consolidation has a positive impact on learning new tasks, it is imperative to maintain model stability to avoid performance degradation of previous tasks. Figure 4 (right) presents the effect of semantic information consolidation on the stability of SPARC. As can be seen, a faster information aggregation leads to lower stability and consequently higher forgetting. On the other hand, no information aggregation can be detrimental when tasks in a sequence are completely different than the first task. Therefore, slow information aggregation coupled with higher value of $\alpha$ leads to a better trade-off between performance and stability of SPARC.

Due to space limitations, we provide additional analysis such as performance of competing approaches with SPARC-like backbone in Section D.2, performance evaluation on ImageNet subset in Section D.3, task-wise performance in Section E.1, task-recency bias in Section E.2, and performance under longer task sequences in Section E.3 in Appendix.

## 5    LIMITATIONS AND FUTURE WORK

We proposed SPARC, a simple rehearsal-free, parameter isolation approach for mitigating catastrophic forgetting in CL devoid of full model surrogates. However, SPARC suffers from number of shortcomings. Firstly, we assume the knowledge of task boundary information to switch between task-specific sub-networks during training. However, this information is not always available in real-world settings. Secondly, SPARC does not take into account the difficulty of each task when allocating learnable task-specific parameters. In cases where current task is extremely difficult or overly easy, static allocation of resources is either insufficient or results in over-parameterization. Furthermore, SPARC grows in size, although more modestly than its peers, with the number of tasks. In longer task sequences consisting of unlimited number of tasks, SPARC grows way beyond other rehearsal-based and weight regularization counterparts. As a future work, task-similarity based weight re-use with more nuanced forward transfer coupled with dynamic resource allocation will further augment SPARC in its endeavour to be a real-world continual learner. Finally, SPARC's current design is specifically optimized for CNN architectures, and its applicability to other model types, such as vision transformers, remains to be explored. Future research will focus on extending SPARC's compact working and semantic memory framework to these architectures, enabling a broader evaluation of its effectiveness and enhancing its adaptability to diverse real-world scenarios.

## 6    CONCLUSION

We introduced SPARC, a rehearsal-free parameter isolation approach for continual learning that operates without the need for full model surrogates. SPARC's design is both simple and efficient, leveraging parameter-efficient task-specific working memories and task-agnostic semantic memory to effectively capture and consolidate information across tasks. Inspired by the Complementary Learning Systems (CLS) theory, this combination of specialized memories allows SPARC to function as an efficient continual learner. Additionally, SPARC incorporates a straightforward weight re-normalization technique in the classification layer to address task-specific biases, ensuring a balanced performance across tasks. While SPARC grows incrementally with each new task, it does so at a slower rate compared to existing methods, maintaining scalability even over long task sequences. Our extensive experimental analysis demonstrates that SPARC achieves comparable or superior performance to rehearsal-based methods across various continual learning benchmarks, all while significantly reducing memory and computational overhead. As a future work, we endeavour to further enhance SPARC with dynamic resource allocation and explore its potential for out-of-distribution generalization, further improving its adaptability and robustness in real-world scenarios.

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

## A   ADDITIONAL INFORMATION

### A.1   TRAINING AND INFERENCE IN CLASS-IL

SPARC is a parameter-isolation continual learning (CL) approach designed to mitigate catastrophic forgetting. It maintains a separate sub-network for each task. During training, as illustrated in Figure 5 (left), a new sub-network is instantiated and trained based on the objective outlined in Eqn. 6 whenever a new task is encountered. After training on a specific task, the corresponding sub-network (i.e., task-specific parameters) remains frozen, while the task-agnostic parameters are updated using an exponential moving average, as described in Eqn. 4. Notably, the CL model is trained simultaneously on both Class-IL and Task-IL settings, as their training regimes are identical. During inference in the Task-IL setting, the appropriate task-specific sub-network is selected based on the given task ID, and its output is inferred for maximum activation. However, inference in the Class-IL setting (see Figure 5 (right)) is more complex, as no task ID is available. In this case, each test image passes through every sub-network, and their respective classifier outputs are concatenated. Since each task-specific sub-network is trained independently and the activation magnitudes produced can be imbalanced, the performance in the Class-IL setting often lags significantly behind that in the Task-IL setting.

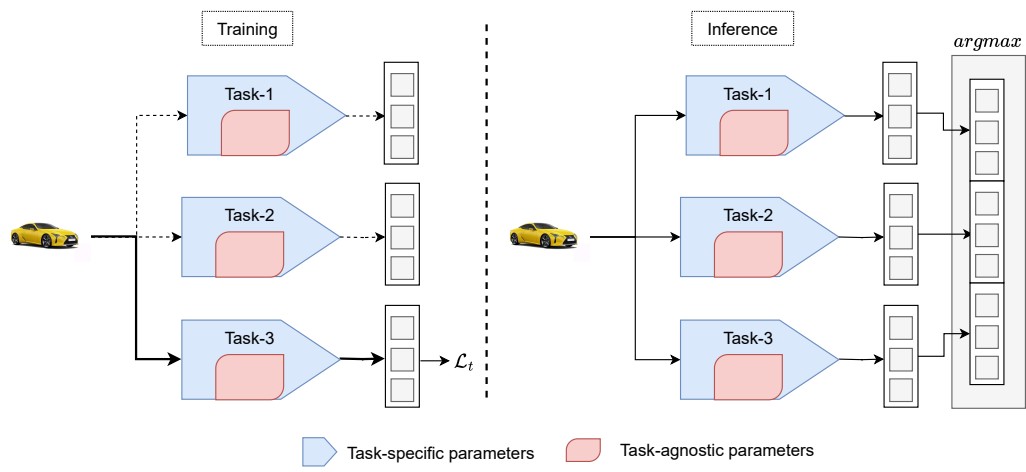

Figure 5: Depiction of training and inference regimes in SPARC in Class-IL setting.

### A.2   PROMINENCE OF WEIGHT RE-NORMALIZATION IN SPARC

Continual learning approaches are prone to task recency bias - the tendency of a CL model to be biased towards classes from the most recent tasks (Masana et al., 2022). Specifically, the model sees only a few or no samples from the old tasks while aplenty from the most recent task, leading to decisions biased towards new classes and the confusion among old classes. While task recency bias is less of a concern in parameter isolation methods due to their modular design and lack of reliance on experience rehearsal, task-specific biases can still emerge, particularly in Class-IL settings. Since each task-specific sub-network is trained independently of all other tasks, including the final fully connected classification layer, the activation magnitudes produced by each sub-network for its respective task can become imbalanced. If one sub-network structurally produces higher output activations, it might also map images from other tasks to disproportionately high activations. This could result in these activations exceeding the activation of the output neuron corresponding to the correct class.

Several approaches have been proposed to address the problem of task recency bias in CL. Since rehearsal-based approaches are more prone to this problem, the solutions also entail bias correction using exemplars stored in the memory buffer. Wu et al. (2019) proposed to learn a linear model on top of trained classifier to reduce the forgetting. Mai et al. (2021) proposed to replace softmax-classifier with nearest mean classifier. In addition, the authors also proposed a supervised contrastive replay to

explicitly encourage samples from the same class to cluster tightly in embedding space while pushing those of different classes further apart during replay-based training. However, the aforementioned approaches require a memory buffer and are not generalizable to parameter-isolation approaches. Zhao et al. (2020) proposed weight alignment that corrects the biased weights in the classification layer after normal training process. Weight Alignment makes full use of the information contained in the trained model and corrects the biased weights in the classification layer without needing a validation set. Similarly, weight re-normalization proposed in this paper scales the classifier weights and biases of each task-specific sub-network based on the maximum activation (after removing outliers) as detailed in Section 3.3. Figure 6 (left) depicts the impact of weight re-normalization on $L_2$-normed classifier weights on SPARC. As can be seen, the weight re-normalization reduces the variance in the weight magnitudes and effectively reduces average magnitude across tasks. We also provide a comparison with weight alignment in SPARC in Figure 6 (right). Weight re-normalization is quite effective in SPARC with far more even distribution of accuracies across tasks. Weight re-normalization also achieves a slight improvement in performance compared to weight alignment. We attribute the success of weight re-normalization to handling of activation during the normalization process: As bias in predictions are a consequence of weights and activations, we speculate that weight alignment strategy that accounts for both is more effective than comparative approaches.

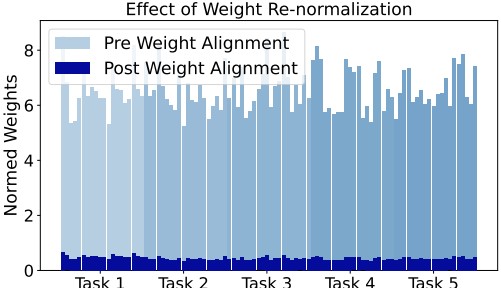 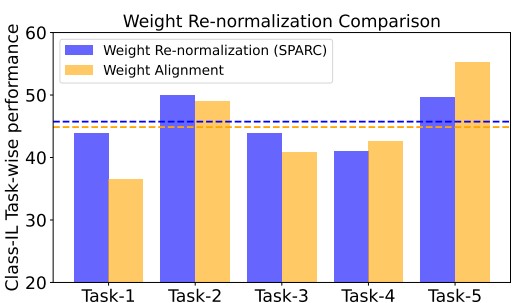

Figure 6: (Left) L2-Normed weights prior to and after weight re-normalization for each task in Seq-CIFAR100-5T. (Right) Comparison of weight re-normalization against weight alignment in SPARC.

## A.3 COMPARISON WITH NON-EXEMPLAR CLASS-IL APPROACHES

Non-Exemplar Class-Incremental Learning (NECIL) is an challenging benchmark designed to address catastrophic forgetting in Class-IL without the need for experience rehearsal. NECIL is particularly advantageous in situations where data confidentiality is paramount due to privacy or security concerns, and where the lifespan of data storage is restricted (Zhai et al., 2024). Early contributions to this field include Learning without Forgetting (LwF) (Li & Hoiem, 2017) and Elastic Weight Consolidation (EWC) (Kirkpatrick et al., 2017), both of which utilize regularization techniques to mitigate the effects of catastrophic forgetting. More recent methods, such as PASS (Zhu et al., 2021b) and IL2A (Zhu et al., 2021a), have enhanced NECIL by generating prototypes for previous classes without needing to retain the original images. The Self-Supervised representation expansion (SSRE) (Zhu et al., 2022) introduced a reparameterization method balancing old and new knowledge, and self-training leverages external data as an alternative for NECIL Additionally, FeTrIL (Petit et al., 2023) proposed a framework that integrates a fixed feature extractor with a pseudo-feature generator using geometric transformations to achieve a better stability-plasticity balance. Recent advancements, such as PRAKA (Shi & Ye, 2023) and PKSPR (Zhai et al., 2024), have further leveraged prototypes to alleviate forgetting in class-incremental settings. ADP (Goswami et al., 2024) proposed adversarial drift compensation technique to estimate semantic drift and resurrect old class prototypes in the new feature space.

NECIL presents a significant challenge as it prohibits the storage of any samples from previous tasks while requiring effective performance in a Class-IL setting. SPARC also falls under NECIL benchmark as it is devoid of any experience rehearsal. Table 6 presents a comparison of NECIL approaches in two Class-IL settings. In addition, we also report model size in terms of number of

Table 6: Average incremental Top-1% accuracy (denoted by Acc) of different non-exemplar Class-IL approaches on Seq-CIFAR100 (5T) and Seq-TinyImageNet (10T). Params (M) represents model size in terms of the number of parameters. Efficiency represents the ratio of average incremental accuracy over the number of parameters in millions. The best results are marked in bold.

| Method | #Params (M) ↓ | Seq-CIFAR100 (5T) | | Seq-TinyImageNet (10T) | |
|---|---|---|---|---|---|
| | | Acc ↑ | Efficiency ↑ | Acc ↑ | Efficiency ↑ |
| EWC (Kirkpatrick et al., 2017) | 14.50 | 16.04 | 1.10 | 15.77 | 1.08 |
| MUC (Liu et al., 2020) | 14.50 | 49.42 | 3.40 | 26.61 | 1.83 |
| SSRE (Zhu et al., 2022) | 19.40 | 65.80 | 3.39 | 48.93 | 2.52 |
| IL2A (Zhu et al., 2021a) | 14.50 | 63.22 | 4.36 | 36.14 | 2.49 |
| PASS (Zhu et al., 2021b) | 14.50 | 63.47 | 4.37 | 47.29 | 3.26 |
| PRAKA (Shi & Ye, 2023) | 22.67 | **70.20** | 3.09 | 52.61 | 2.32 |
| FeTrIL (Petit et al., 2023) | 11.27 | 66.30 | 5.88 | **53.10** | 4.71 |
| ADC (Goswami et al., 2024) | 22.67 | 59.17 | 2.61 | 50.94 | 2.24 |
| PKSPR (Zhai et al., 2024) | 9.30 | 68.17 | 7.33 | 52.72 | 5.66 |
| **SPARC** | **1.04** | 63.58 | **61.14** | 44.03 | **23.12** |

parameters in millions and efficiency computed as the ratio of average incremental accuracy (top-1%) in Class-IL over model size. Owing to larger model sizes, other approaches perform better than SPARC while foregoing the efficiency. On the other hand, SPARC has minimal footprint in terms of model size among all compatriots. In addition, SPARC is the most efficient across both Class-IL settings by a huge margin. As noted earlier in Table 2, the performance of SPARC can simply be boosted further by increasing either the width or depth or both, at the cost of efficiency.

## A.4 Comparison with PEFT techniques in Vision Transformers

Parameter Efficient Fine-Tuning (PEFT) techniques that have emerged in the space of Large Large Language Models (LLMs) have played a crucial role in enhancing model efficiency while maintaining low memory and computational costs for fine-tuning. Notable innovations like Adapter modules (Houlsby et al., 2019), Prompt Tuning (Brown, 2020), BitFit (Zaken et al., 2021), Low Rank Adaptation (LoRA) (Hu et al., 2021), and DoRA (Liu et al., 2024) have showcased the benefits of selective fine-tuning, effectively balancing model generalization with increased adaptability. In the context of CL, various strategies utilize PEFT techniques to address catastrophic forgetting in vision transformers. Approaches such as L2P (Wang et al., 2022e) and DualPrompt (Wang et al., 2022d) employ task-specific prompts to support the acquisition of new tasks while safeguarding previously learned knowledge. Extensions like S-Prompt (Wang et al., 2022a) and CODA-Prompt (Smith et al., 2023) leverage structural prompts to delineate discriminative relationships and apply Schmidt orthogonalization, respectively. EASE (Zhou et al., 2024) develops task-specific subspaces by utilizing lightweight adapter modules tailored for each new task, while InfLoRA (Liang & Li, 2024) introduces a small set of parameters to reconfigure the pre-trained weights, demonstrating that fine-tuning these new parameters can achieve results comparable to fine-tuning the original pre-trained weights within a defined subspace. Despite the relative success of these methods, they are predominantly designed for pre-trained vision transformers, which necessitates the availability of a pre-trained model. Consequently, in situations where resource efficiency is paramount, the applicability of these techniques becomes limited.

On the other hand, SPARC is a parameter-efficient parameter isolation approach tailored for CNNs. Unlike these approaches, SPARC does grow, albeit more slowly than its counterparts, with more tasks in the order. Nevertheless, SPARC remains scalable even in the face longer task sequences compared PEFT techniques. Table 7 provides an efficiency comparison of SPARC against PEFT techniques in CL. We duly note that this comparison is not an apple-to-apple comparison since (i) PEFT techniques use a model pre-trained on ImageNet-21k while SPARC is trained from scratch, (ii) The base model ViT-base-16 used in PEFT techniques is far bigger than SPARC backbone, and finally (iii) The baselines and SPARC are tailored for different architectures, vision transformers and CNNs respectively. Having said that, SPARC still shows much higher efficiency compared to PEFT techniques. We also note that the performance of SPARC can be easily boosted by increasing either the width or depth or both at the cost of efficiency.

Table 7: Efficiency comparison of SPARC against PEFT techniques in Seq-CIFAR100 (10T). All baselines use ViT-base-16 with a footprint of approximately 86 Million parameters while SPARC has a footprint of 1.90 Million. Here, Efficiency represents the ratio of average accuracy (top-1%) over all tasks at the end of the training Vs the number of parameters in millions. The best results are marked in bold.

| Method | #Params (M) $\downarrow$ | Seq-CIFAR100 (10T) Efficiency $\uparrow$ |
|---|---|---|
| L2P (Wang et al., 2022e) | $\approx 86$ | 0.98 |
| DualPrompt (Wang et al., 2022d) | $\approx 86$ | 0.99 |
| CODA-Prompt (Smith et al., 2023) | $\approx 86$ | 1.01 |
| EASE (Zhou et al., 2024) | $\approx 86$ | 1.02 |
| InfLoRA (Liang & Li, 2024) | $\approx 86$ | 1.01 |
| **SPARC** | **1.90** | **23.68** |

### A.5 FORGETTING ANALYSIS

We present additional metrics for our experiments to thoroughly assess the performance of SPARC. Table 8 includes further metrics such as forgetting, stability, and plasticity, which offer deeper insights into the model's behavior. It is important to note that most baseline models do not offer these metrics, which limits our ability to make comparisons across these measures. Forgetting gauges the model's capacity to retain knowledge from prior tasks by measuring the average decline in accuracy of a task at the end of continual learning training compared to its initial accuracy; lower forgetting values indicate better retention of knowledge. Stability (S) represents the average accuracy on previously learned tasks at the end of training, showcasing the model's performance on earlier tasks. Conversely, plasticity (P) assesses the model's ability to learn new tasks effectively, determined by the average accuracy of tasks during their initial training. The trade-off is quantified by the formula $(2 \times S \times P)/(P + S)$, which indicates how well the method balances stability and plasticity. Collectively, these metrics provide a comprehensive overview of SPARC's performance, highlighting its strengths and trade-offs in various Class-IL contexts.

In the Seq-CIFAR100 (5T) context, competing methods demonstrate moderate stability and high plasticity, resulting in a suboptimal trade-off between the two. In contrast, SPARC achieves a more advantageous balance, leading to superior knowledge aggregation and retention across tasks. This effectiveness is attributed to the nuanced interplay between working and semantic memory systems within SPARC. Additionally, the design of SPARC allows for efficient, surrogate-free assimilation of CLS theory, thereby enhancing the trade-off between stability and plasticity while maintaining parameter efficiency.

Table 8: SPARC Class-IL performance metrics across datasets.

| Dataset | Method | Accuracy (%) $\uparrow$ | Forgetting (%) $\downarrow$ | Stability (%) $\uparrow$ | Plasticity (%) $\uparrow$ | Trade-off $\uparrow$ |
|---|---|---|---|---|---|---|
| Seq-CIFAR10 (5T) | SPARC | 63.75 | 18.25 | 62.51 | 78.35 | 69.54 |
| Seq-CIFAR100 (5T) | ER | 27.78 | 68.35 | 12.89 | **82.4** | 22.31 |
| | DER++ | 42.74 | 47.35 | 33.15 | 80.62 | 46.98 |
| | LiDER | 42.31 | 43.72 | 34.33 | 77.27 | 47.54 |
| | SPARC | **53.51** | **13.41** | **51.8** | 64.24 | **57.35** |
| Seq-TinyImageNet (10T) | SPARC | 32.38 | 12.58 | 32.55 | 43.71 | 37.31 |

## B RELATED WORKS

### B.1 REHEARSAL BASED METHODS

Continual learning on a sequence of tasks remains a persistent challenge for DNNs due to the catastrophic forgetting of older tasks. Experience-rehearsal (ER) (Ratcliff, 1990; Lin, 1992), which stores and replays a subset of training samples from previous tasks, is one of the earliest approaches devised to mitigate catastrophic forgetting in CL. Several methods build on top of ER: LUCIR (Hou

et al., 2019) proposed to use cosine normalization and inter-class separation, to mitigate the adverse effects of the class imbalance in Class-IL. DER (Buzzega et al., 2020) combines rehearsal with consistency regularization, aligning the network's logits over the course of optimization to maintain consistency with its past behavior. The authors also propose an extension to DER, termed DER++, which promotes logits consistency as well as encouraging the network to more accurately predict the correct ground truth label. Multiple proposals have been made in conjunction with DER++ to further augment reduction in catastrophic forgetting: LiDER (Bonicelli et al., 2022) proposed a Lipschitz-driven rehearsal, a surrogate objective that reduces overfitting on buffered samples and improves generalization for rehearsal-based approaches. ER-ACE (Caccia et al., 2021) introduces a simple adjustment in the cross-entropy loss to nudge learned representations to be more robust to new future classes. The goal is to avoid drastic representation drift that can negatively affect the performance of a continual learning model. The implementation of ER-ACE is efficient in both memory and compute. Although these approaches reduce catastrophic forgetting by a large extent, their performance is tightly tied to the buffer size: lower buffer size leads to overfitting while large buffer size is not tenable in memory constrained devices. GCR (Tiwari et al., 2022) selects a subset of past data that best approximates the gradient of the entire dataset seen so far and combines this with logit distillation similar to DER++ and contrastive learning.

On the other hand, several approaches employ one or more full model surrogates to improve forgetting: CLS-ER (Arani et al., 2022) uses two additional models to separate learning from memory consolidation. TAMiL (Bhat et al., 2023) entails a single additional model along with as many task-specific attention modules as the number of tasks. OCD-Net (Li et al., 2022) employs a teacher-student framework, where the teacher model aids the student model in consolidating knowledge. SCoMMER (Sarfraz et al., 2022b) and TriRE (Vijayan et al., 2023) enforce activation sparsity in conjunction with a dropout mechanism, which encourages the model to activate similar units for semantically related inputs while reducing the overlap in activation patterns for semantically unrelated inputs. Additionally, both employ a long-term semantic memory that consolidates the information encoded in the working model. $Co^2L$ (Cha et al., 2021) leverages self-supervised contrastive learning to develop generalizable features across tasks. It also uses a snapshot of the most recent model and a distillation loss to retain learned features from previous tasks, necessitating the storage of one additional model. While these methods bridge the gap between independent and identically distributed (iid) and non-iid training, they rely on one or more surrogate models to mitigate forgetting.

### B.2 WEIGHT REGULARIZATION METHODS

As models accumulate knowledge through training on data, this knowledge becomes embedded in the network's weights. Weight regularization methods address catastrophic forgetting by imposing constraints on the updates of these weights, often through modifying the objective function. The goal is to preserve essential knowledge from previously learned tasks within the model parameters while remaining adaptable to acquire new knowledge. One of the earliest weight regularization approaches is Elastic Weight Consolidation (EWC) (Kirkpatrick et al., 2017) selectively regulates the plasticity of neural network parameters crucial for previously acquired knowledge. EWC employs Fisher Information Matrix (FIM), which signifies the importance of each parameter regarding prior tasks. Learning without Forgetting (LwF) (Li & Hoiem, 2017) leverages the knowledge embedded in the network's parameters to approximate its performance on past tasks. During training on a new task, LwF promotes the network's response consistency by recording the initial logits for the new task's samples and adding a loss term that encourages the model to align its current logits with these recorded logits, thereby preserving its previous knowledge. MUC (Liu et al., 2020) builds on top of LwF and integrates an ensemble of auxiliary classifiers to effectively estimate regularization constraints. Synaptic Intelligence (SI) (Zenke et al., 2017) quantifies the synaptic contribution of each parameter to the loss reduction over the course of a task. This quantification determines a cost for modifying each parameter, effectively measuring its importance to learned tasks. This measure is then used to penalize changes to these parameters in future learning. Gradient Projection Memory (GPM) (Saha et al., 2021) aims to conserve knowledge from previous tasks by taking gradient steps orthogonal to the gradient sub-spaces important for past tasks. Building on GPM, (Abbasi et al., 2022) introduced a method combining GPM with sparsity through k-winner activations with Heterogeneous Dropout (HD). HD encourages the network to utilize distinct activation patterns for different tasks, promoting task-specific knowledge preservation. ALASSO (Park et al., 2019) introduced a CL framework that involves overestimating the unobserved aspect of a loss function for

the current task and approximating the loss using an asymmetric quadratic function. This approach enables a reliable estimation of loss even in the absence of training data from previous tasks. UCB (Ebrahimi et al., 2020) hinges on an assumption that uncertainty is a natural way to identify what to remember and what to change as we continually learn, and thus mitigate catastrophic forgetting. To this end, UCB proposed a method that utilizes Bayesian neural networks to adapt the learning rate of individual parameters based on the inherent measure of uncertainty. Inspired by the multi-level human memory system, BMKP (Sun et al., 2023) proposed a bi-level-memory framework with a representation compaction regularizer designed to encourage the working memory to reuse previously learned knowledge, which enhances both the memory efficiency and the performance.

While weight regularization methods generally offer the advantage of not requiring a memory buffer, reducing memory overhead, and speeding up training by eliminating the need to retrain on previous task data, they often face challenges such as limited capacity for adapting to new knowledge, the introduction of additional hyperparameters, and the difficulty in balancing the stability-plasticity dilemma. Overly strict constraints on updating parameters important for previous tasks can lead to limited forward information transfer between tasks, obstructing the development of efficient and generalizable representations.

## B.3 PARAMETER ISOLATION METHODS

It is widely recognized that the brain, particularly the neocortex, exhibits a high degree of sparsity. This sparsity is manifested through various mechanisms. Firstly, the inter-connectivity between neurons is sparse. In-depth anatomical studies reveal that cortical pyramidal neurons receive relatively few excitatory inputs from neighboring neurons (Markram et al., 2015). The proportion of local area connections seems to be less than $5\%$ (Holmgren et al., 2003), in stark contrast to a fully connected dense network. In addition to sparse connectivity, several studies indicate that only a small percentage of neurons become active in response to sensory stimuli (Attwell & Laughlin, 2001; Barth & Poulet, 2012). Furthermore, the grid cells in the brain's entorhinal cortex enable the brain to encode spatial information efficiently, fostering sparsity by selectively activating a small percentage of cells in response to specific locations or stimuli, thereby optimizing neural resources for spatial cognition and navigation. The pervasiveness of sparsity in the neocortex, associated with its capacity to generate meaningful representations, make predictions, and detect surprises and anomalies, underscores its fundamental role in enhancing efficiency and functionality.

Inspired by how brain functions in a sparse manner, several approaches attempted dynamic sparsity within fixed model capacity in CL. Motivated by persistent dendritic spines, approaches such as CLNP (Golkar et al., 2019), NISPA (Gurbuz & Dovrolis, 2022), PackNet (Mallya & Lazebnik, 2018), and PAE (Hung et al., 2019b)) proposed dynamic sparse networks based on neuronal model sparsification with fixed model capacity. NISPA forms task-specific sparse stable paths to preserve learned knowledge from older tasks. As a consequence, NISPA entails as many masks as the number of tasks. WSN (Kang et al., 2022) reduces the overhead by encoding masks into one N-bit binary digit mask, then compressing using Huffman coding for a sub-linear increase in network capacity with respect to the number of tasks. SparCL (Wang et al., 2022c) entails a task-aware dynamic masking strategy that dynamically removes less important weights and grows back unused weights for stronger representation power periodically by maintaining a single binary weight mask throughout the CL process. However, methods such as NISPA and WSN store several model surrogates to mask out different parts of the network for different tasks, thereby resulting in massive overhead.

Several parameter-isolation approaches (Rusu et al., 2016) significantly grow beyond fixed model capacity to reduce catastrophic forgetting by substantially reducing overlap of parameters associated with each task. As a consequence, the number of model surrogates explodes in longer task sequences, rendering them unscalable in real-world applications. Progressive Neural Networks (PNNs (Rusu et al., 2016)) create a new sub-network for each task, incorporating lateral connections to previously learned frozen models. DEN (Yoon et al., 2018) introduced a dynamically expandable network by incorporating selective retraining, network expansion with group sparsity regularization, and neuron duplication. Likewise, CPG (Hung et al., 2019a) presented an iterative method that includes pruning previous task weights and gradually expanding the network while reusing crucial weights from previous tasks.

Table 9: List of methods used for benchmark comparison on Class-IL or Task-IL.

| Rehearsal Based Method | | | |
|---|---|---|---|
| DER++ | (Buzzega et al., 2020) | CLS-ER | (Arani et al., 2022) |
| ER-ACE | (Caccia et al., 2021) | OCDNet | (Li et al., 2022) |
| $Co^2L$ | (Cha et al., 2021) | TAMiL | (Bhat et al., 2023) |
| GCR | (Tiwari et al., 2022) | TriRE | (Vijayan et al., 2023) |
| LUCIR | (Hou et al., 2019) | | |
| **Weight Regularization Method** | | | |
| oEWC | (Kirkpatrick et al., 2017) | SI | (Zenke et al., 2017) |
| LwF | (Li & Hoiem, 2017) | MUC | (Liu et al., 2020) |
| ALASSO | (Park et al., 2019) | UCB | (Ebrahimi et al., 2020) |
| BMKP | (Sun et al., 2023) | | |
| **Parameter Isolation Method** | | | |
| PNNs | (Rusu et al., 2016) | PackNet | (Mallya & Lazebnik, 2018) |
| NISPA | (Gurbuz & Dovrolis, 2022) | SparCL-EWC | (Wang et al., 2022c) |
| CLNP | (Golkar et al., 2019) | PAE | (Hung et al., 2019b) |
| CPG | (Hung et al., 2019a) | | |

## C    DEPTH-WISE SEPARABLE CONVOLUTIONS

A Depth-wise Separable Convolutional (DSC) layer, often referred to as separable convolution in the literature, comprises a depth-wise convolution followed by a point-wise convolution operation, without any non-linearity between them. In contrast to a traditional convolutional layer, which applies a convolutional operation over the entire input volume by combining spatial and cross-channel convolutions in a single step, the depth-wise convolution performs spatial convolution independently on each input channel (Howard et al., 2017). Subsequently, the point-wise convolution (i.e., a 1x1 convolution) projects the outputs of the depth-wise convolution onto a new channel space. This separation reduces the number of parameters and enhances computational efficiency compared to a standard convolutional layer. While a traditional convolutional layer with $c_1$ input channels and $c_2$ output channels, and kernel dimensions $h$ by $w$, uses $h \times w \times c_1 \times c_2$ parameters, a depth-wise convolutional layer can significantly reduces this by using $h \times w \times c_1 + c_1 \times c_2$ parameters (Guo et al., 2019).

The concept of depth-wise separable convolution was first introduced by (Sifre & Mallat, 2014), in a paper on rigid-motion scattering for texture classification, and later applied to AlexNet (Krizhevsky et al., 2012), resulting in improved accuracy, enhanced convergence speed, and reduced model size. This technique was further exploited by (Howard et al., 2017; Sandler et al., 2019) in the development of MobileNet, a lightweight deep neural network (DNN) designed for mobile and embedded visual applications, significantly advancing the field of efficient neural network design. (Chollet, 2017) proposes the Xception architecture and interprets Inception (Szegedy et al., 2015) modules as an intermediate step between regular convolutional layers and DSC layers. By replacing Inception modules with DSC layers, Xception achieves superior performance due to more efficient parameter use facilitated by depth-wise separable convolutions. Moreover, (Guo et al., 2019) demonstrated that DSC layers can enhance networks tasked with learning multiple visual domains. Their research posits that different visual domains possess domain-specific spatial correlations but share cross-channel correlations, thus benefiting from DSC layers.

As the CL paradigm is well-suited for learning across multiple visual domains, we argue that DSC layers are ideally suited for this setting in machine learning. Furthermore, DSC layers enable more efficient computation and a reduced number of parameters, making the architecture highly scalable as the number of tasks increases, which is a crucial aspect of CL.

Table 10: Top-1 accuracy (%) of different rehearsal-based CL models in Class-IL and Task-IL scenarios with buffer size 500. The best results are marked in bold.

| Method | #Params (M) | # of $\mathcal{F}$ and $\mathcal{B}$ | Seq-CIFAR10 (5T) | | Seq-CIFAR100 (5T) | | Seq-TinyImageNet (10T) | |
|---|---|---|---|---|---|---|---|---|
| | | | Class-IL | Task-IL | Class-IL | Task-IL | Class-IL | Task-IL |
| JOINT | 11.23 | $1\mathcal{F}$ , $1\mathcal{B}$ | 92.20 $_{\pm 0.15}$ | 98.31 $_{\pm 0.12}$ | 70.56 $_{\pm 0.28}$ | 86.19 $_{\pm 0.43}$ | 59.99 $_{\pm 0.19}$ | 82.04 $_{\pm 0.10}$ |
| SGD | 11.23 | $1\mathcal{F}$ , $1\mathcal{B}$ | 19.62 $_{\pm 0.05}$ | 61.02 $_{\pm 3.33}$ | 17.49 $_{\pm 0.28}$ | 40.46 $_{\pm 0.99}$ | 7.92 $_{\pm 0.26}$ | 18.31 $_{\pm 0.68}$ |
| ER | 11.23 | $1\mathcal{F}$ , $1\mathcal{B}$ | 57.74 $_{\pm 0.27}$ | 93.61 $_{\pm 0.27}$ | 28.02 $_{\pm 0.31}$ | 68.23 $_{\pm 0.17}$ | 9.99 $_{\pm 0.29}$ | 48.64 $_{\pm 0.46}$ |
| DER++ | 11.23 | $2\mathcal{F}$ , $1\mathcal{B}$ | 72.70 $_{\pm 1.36}$ | 93.88 $_{\pm 0.50}$ | 41.40 $_{\pm 0.96}$ | 70.61 $_{\pm 0.08}$ | 19.38 $_{\pm 1.41}$ | 51.91 $_{\pm 0.68}$ |
| ER-ACE | 11.23 | $1\mathcal{F}$ , $1\mathcal{B}$ | 68.45 $_{\pm 1.78}$ | 93.47 $_{\pm 1.00}$ | 40.67 $_{\pm 0.06}$ | 66.45 $_{\pm 0.71}$ | 17.73 $_{\pm 0.56}$ | 49.99 $_{\pm 1.51}$ |
| Co$^2$L | 22.67 | $4\mathcal{F}$ , $1\mathcal{B}$ | 74.26 $_{\pm 0.77}$ | 95.90 $_{\pm 0.26}$ | 39.21 $_{\pm 0.39}$ | 62.98 $_{\pm 0.58}$ | 20.12 $_{\pm 0.42}$ | 53.04 $_{\pm 0.69}$ |
| GCR | 11.23 | $1\mathcal{F}$ , $1\mathcal{B}$ | 74.69 $_{\pm 0.85}$ | 94.44 $_{\pm 0.32}$ | 45.91 $_{\pm 1.30}$ | 71.64 $_{\pm 2.10}$ | 19.66 $_{\pm 0.68}$ | 52.99 $_{\pm 0.89}$ |
| CLS-ER | 33.69 | $3\mathcal{F}$ , $1\mathcal{B}$ | 75.22 $_{\pm 0.71}$ | 94.94 $_{\pm 0.53}$ | 51.40 $_{\pm 1.00}$ | 78.12 $_{\pm 0.24}$ | 31.03 $_{\pm 0.56}$ | 60.41 $_{\pm 0.50}$ |
| OCDNet | 22.46 | $2\mathcal{F}$ , $1\mathcal{B}$ | 80.64 $_{\pm 0.77}$ | 96.57 $_{\pm 0.07}$ | 54.13 $_{\pm 0.36}$ | 78.51 $_{\pm 0.24}$ | 26.09 $_{\pm 0.28}$ | 64.76 $_{\pm 0.29}$ |
| TAMiL | 23.10 | $2\mathcal{F}$ , $1\mathcal{B}$ | 74.45 $_{\pm 0.27}$ | 94.61 $_{\pm 0.19}$ | 50.11 $_{\pm 0.34}$ | 76.38 $_{\pm 0.30}$ | 28.48 $_{\pm 1.50}$ | 64.42 $_{\pm 0.27}$ |
| TriRE | 100.98 | $2\mathcal{F}$ , $1\mathcal{B}$ | 68.17 $_{\pm 0.33}$ | 92.45 $_{\pm 0.18}$ | 43.91 $_{\pm 0.18}$ | 71.66 $_{\pm 0.44}$ | 20.14 $_{\pm 0.19}$ | 55.95 $_{\pm 0.78}$ |
| **SPARC** | **1.04** | $1\mathcal{F}$ , $1\mathcal{B}$ | 61.22 $_{\pm 4.81}$ | 95.76 $_{\pm 0.21}$ | 49.03 $_{\pm 0.05}$ | 75.52 $_{\pm 0.11}$ | **32.29** $_{\pm 0.01}$ | **65.66** $_{\pm 0.01}$ |

Table 11: Performance of competing approaches with DSCs and same number of filters as SPARC on Seq-CIFAR100 with 5 tasks. Competing methods employ a buffer of size 500.

| Method | #Params (M) | #Filters per task | Class-IL | Task-IL |
|---|---|---|---|---|
| ER | 7.85 | [160, 320, 640, 1280] | 22.80 $_{\pm 0.87}$ | 60.64 $_{\pm 0.73}$ |
| DER++ | 7.85 | [160, 320, 640, 1280] | 27.78 $_{\pm 1.27}$ | 65.95 $_{\pm 1.87}$ |
| SPARC | **1.04** | [32, 64, 128, 256] | **49.03** $_{\pm 0.05}$ | **75.52** $_{\pm 0.11}$ |

# D ADDITIONAL EXPERIMENTS

## D.1 ADDITIONAL RESULTS WITH BUFFER SIZE 500

In Table 1, we compare and contrast SPARC with several rehearsal-based method with buffer size 200. Due to space limitations, Table 10 provides additional results pertaining to rehearsal-based methods with buffer size 500. As can be seen between Tables 10 and 1, a larger buffer size greatly improves performance across tasks for rehearsal-based methods there by resulting in reduced forgetting. On the other hand, except for Seq-CIFAR10, SPARC outperforms every rehearsal-based method in buffer size 200 without experience rehearsal. SPARC is also quite competitive in buffer size 500 category without the use of experience rehearsal and full model surrogates.

## D.2 PERFORMANCE OF COMPETING APPROACHES WITH SPARC-LIKE BACKBONE

We investigate the impact of parameter-efficient DSCs within the backbone of competing methods. To this end, we employ the same backbone described in Figure 1: Both SPARC and competing methods are equipped with the same number of filters and are equally wide. Competing methods are trained with a buffer size of 500 on Seq-CIFAR100 with 5 tasks. As can be seen in Table 11, parameter isolation between tasks effectively reduces the number of learnable parameters even with the same number of filters. Secondly, SPARC with a fraction of parameters outperforms competing methods without explicit experience rehearsal. Thus, the performance of SPARC cannot be attributed solely to DSCs. Its complex conjugation of working and semantic memories that enable SPARC to be compact, scalable, and surrogate-free in CL.

## D.3 PERFORMANCE ON IMAGENET SUBSET

We present the performance evaluation of SPARC on ImageNet subset in Table 12: Seq-MiniImageNet. Since the majority of the baselines in Table 1 do not report results on these datasets due to computational constraints, we report results on ER and DER++. The competing approaches employ a buffer size of 100 while SPARC is rehearsal-free. As can be seen, SPARC outperforms the baselines without relying on experience rehearsal even in Seq-MiniImageNet.

Table 12: Top-1 accuracy (%) of different CL models in Class-IL and Task-IL scenarios in Seq-MiniImageNet with 20 task. The best results are marked in bold.

| Method | #Params (M) | # of $\mathcal{F}$ and $\mathcal{B}$ | Class-IL | Task-IL |
|---|---|---|---|---|
| ER | 11.23 | $1\mathcal{F}$ , $1\mathcal{B}$ | 22.64 ±0.50 | 53.43 ±1.18 |
| DER++ | 11.23 | $2\mathcal{F}$ , $1\mathcal{B}$ | 23.86 ±0.62 | 59.80 ±1.51 |
| **SPARC** | **3.62** | $1\mathcal{F}$ , $1\mathcal{B}$ | **27.20** ±0.20 | **80.04** ±0.26 |

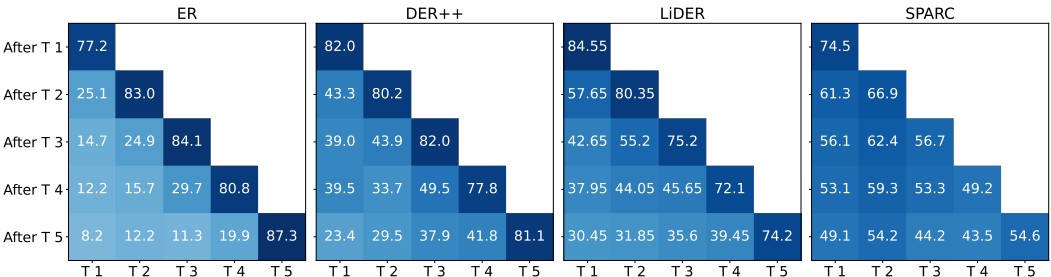

Figure 7: Comparison of task-wise performance of several methods on Seq-CIFAR100 divided into 5 tasks. SPARC shows balanced performance on all 5 tasks after the completion of the training phase.

# E  FURTHER ANALYSIS

## E.1  TASK-WISE PERFORMANCE

We present the final accuracy after learning all tasks in both Class-IL and Task-IL scenarios in Table 1 and 10. Furthermore, in Figure 7, we analyze the task-wise performance of various CL models in Class-IL trained on Seq-CIFAR100 with a buffer size of 500 for 5 tasks. As can be seen, ER and DER++ produce skewed performance, while SPARC produces a well-distributed performance across tasks. We attribute this behavior to weight re-normalization as it corrects weight magnitude disparity after every task training resulting in lower task-specific biases.

## E.2  TASK-RECENCY BIAS

Task-recency bias in CL models is characterized by their predisposition to perform better on tasks that have been encountered more recently compared to those learned earlier in the training process (Masana et al., 2022). Task recency bias often leads to decisions that favor newer classes, leading to ambiguity between older classes (Bhat et al., 2022). Balanced training, nearest mean exemplar classification, loss weighting, and reducing task imbalance are some of the approaches to reduce task recency bias in CL. Following (Masana et al., 2022; Arani et al., 2022; Sarfraz et al., 2022a), Figure 8 presents the normalized task probabilities for different approaches, including SPARC. For any given trained model, normalized probabilities are obtained by presenting the model with a balanced test set and recording the number of classifications for each task. Finally, the counts are divided by the total number of test samples to find the normalized probabilities. As can be seen, SPARC obtains evenly distributed task probabilities compared to competing approaches, effectively eliminating any noticeable task-recency bias.

## E.3  CAPACITY SATURATION UNDER LONGER TASK SEQUENCES

Parameter isolation approaches, including SPARC, offer maximum stability by fixing all or a subset of parameters belonging to previous tasks. However, parameter isolation approaches within a fixed capacity suffer from capacity saturation, i.e., there is not enough free parameters to capture new tasks resulting in reduced or no plasticity (De Lange et al., 2021). In SPARC, however, we assume the knowledge of the number of tasks beforehand and equally distribute total capacity among all tasks. We conduct experiments on Seq-CIFAR100 with 5, 10, 20, and 50 tasks to understand how SPARC

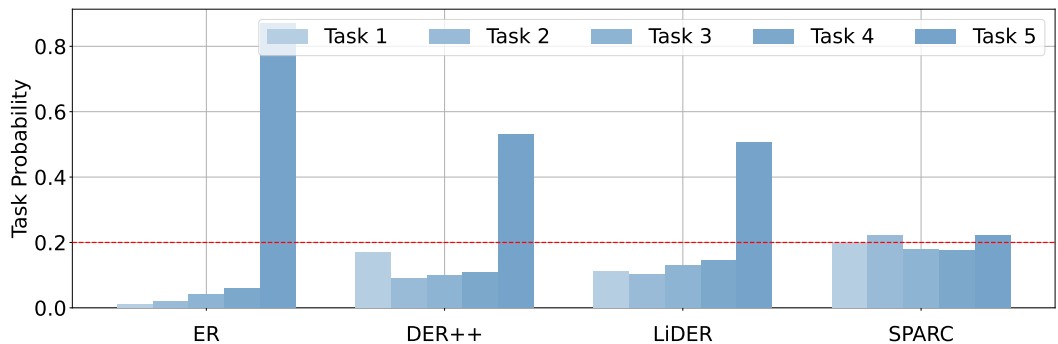

Figure 8: Comparison of task-probabilities on Seq-CIFAR100 with 5 tasks. SPARC achieves a significantly better balance between tasks compared to other methods.

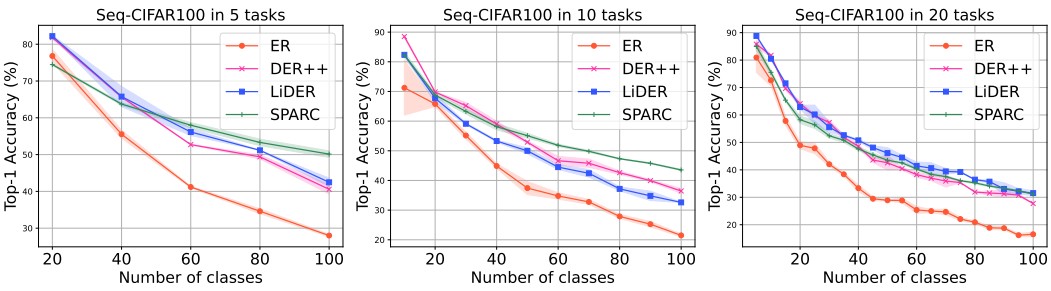

Figure 9: Top-1 Accuracy (%) on Seq-CIFAR100 with 5, 10, and 20 tasks respectively. The graphs compare the performance of SPARC to ER and DER++ across varying numbers of tasks. SPARC consistently maintains higher accuracy than ER and DER++ as the number of tasks increase.

behaves in longer task sequences. Figure 9 shows the average accuracy after each task for each of these experiments. Although SPARC grows in size with every new task in the sequence, it does so more modestly compared to other parameter isolation methods. For instance, SPARC uses 3.62 Million parameters while ER and DER++ utilize 11.23 Million parameters to learn the same 20 tasks in Seq-CIFAR100. Moreover, they utilize experience rehearsal on top to combat forgetting. On the other hand, SPARC produces a better performance with quite fewer parameters, without experience rehearsal and full model surrogates.

## F  IMPLEMENTATION DETAILS

### F.1  DATASETS AND SETTINGS

Class-Incremental Learning (Class-IL) and Task-incremental learning (Task-IL) are two prominent paradigms for effectively evaluating different approaches in CL. In Class-IL, the model is presented with a series of tasks featuring non-overlapping classes. The primary challenge here is to correctly classify new instances from all classes seen thus far, necessitating the model to not only learn to discriminate within each specific task but also to distinguish between different tasks. In Task-IL, the model is provided with a task identifier during both training and inference, effectively eliminating the need for the model to differentiate between tasks, thus allowing it to concentrate solely on discrimination within the given task.

Following recent research trends in CL, we create Seq-CIFAR10, Seq-CIFAR100, and Seq-TinyImageNet by dividing CIFAR10 (Krizhevsky et al., 2009), CIFAR100 (Krizhevsky et al., 2009), and TinyImageNet (Le & Yang, 2015) into 5, 5, and 10 partitions with 2, 20, and 20 classes per task, respectively. In Seq-CIFAR100, we also experiment with longer task sequences, increasing the

number of tasks to 5, 10, and 20, while correspondingly decreasing the number of classes per task to 20, 10, and 5, respectively (see Appendix E.3). We also employ ImageNet subsets to evaluate SPARC: Seq-MiniImageNet and Seq-ImageNet100. Seq-MiniImageNet (Aljundi et al., 2019) splits full ImageNet classification dataset to 20 disjoint subsets by their labels. The dataset consists of 20 tasks with an overall 100 classes, where each task consists of 1,250 examples in total from 5 classes. On the other hand, Seq-ImageNet100 employs the first 100 classes of ImageNet dataset and divides it among 10 tasks. Seq-ImageNet100 maintains a full image resolution in training, while Seq-MiniImageNet uses a reduced image resolution of 84x84. The training regime for both Class-IL and Task-IL involves training the CL model sequentially on all tasks with or without experience-rehearsal, using reservoir sampling depending on the formulation. This training scheme is consistent for both Class-IL and Task-IL. For comparison with state-of-the-art methods, we report the average accuracies on all tasks seen so far in Class-IL. In Task-IL, we leverage the task identity and mask neurons that do not belong to the prompted task in the linear classifier, following standard practice. For all our experiments we use a single NVIDIA GeForce 8GB GPU to train SPARC on each of the datasets mentioned in Table 1.

**Hyperparameters:** Across all datasets, we use the same set of hyperparameters to show the simplicity and effectiveness of SPARC across scenarios. Specifically, we use a width of 0.5 and a depth of 4 to make SPARC backbone resemble ResNet-18. For each task, we reserve 32, 64, 128, and 256 depth-wise filters per task in layers 1 to 4 respectively. This is a conscious choice to provide a fair comparison with competing approaches. All throughout, we use a learning rate of 5e-3, batch size of 32, EMA $\alpha$ of 0.99 and 50 training epochs per task. The weight re-normalization $\kappa$ is a constant, set to 5 within our experiments. We re-iterate that we do not use any dataset-specific hyperparameter tuning to find the best results.

**Evaluation metrics**: We report two kinds of accuracy metrics throughout this paper: Class-IL and Task-IL accuracy, and incremental accuracy. Class-IL and Task-IL accuracies represent top-1% average accuracy across all tasks after CL training. On the other hand, incremental accuracy is computed as the average accuracy of all incremental phases, including the initial one. As Class-IL and Task-IL accuracies are a common practice in the literature, all our results correspond to this metric except those in Table 3. Since majority of the baselines did not report results on Seq-ImageNet100, we report the incremental accuracy within Table 3 to conform to the baselines.

## F.2 DETAILS ON EXCEPTIONS IN TABLE 1 AND FIGURE 2

While most of the results shown in Table 1, including those for SPARC, have been achieved after 50 training epochs, there are some exceptions. For the more challenging sequential TinyImageNet dataset, ER, DER++, DER w/ SD, GCR, SparCL and the linear classifier of Co²L were trained for twice as many epochs (i.e. 100 epochs). In contrast, IMEX-Reg was trained for only 20 epochs on this dataset, and 50 epochs on the sequential CIFAR-10 and CIFAR-100 datasets. DER++ w/ FPF was only trained for 5 epochs on all three dataset.

