# OpenReview forum: "SPARC: Continual learning beyond experience rehearsal and model surrogates"
_ICLR.cc/2025/Conference — Submitted to ICLR 2025_

### Official Review · Reviewer_fWSr · 2024-10-22

**Soundness:** 2
**Presentation:** 3
**Contribution:** 2
**Rating:** 5
**Confidence:** 4

**Summary:**

This work focuses on continual learning, leveraging task-specific information as memory for cross-task knowledge consolidation. The proposed method eliminates the need for memory-intensive experience rehearsal and model surrogates, while minimizing forgetting. Empirical evaluations on various continual learning tasks demonstrate the method's superior performance in terms of accuracy and parameter efficiency.

**Strengths:**

The discussion on rehearsal memory is well-supported with sufficient related work and analysis. The study of rehearsal-free continual learning methods is important.

**Weaknesses:**

The method still relies on task boundary information for task-specific model learning, and further investigation into this aspect and relevant work is not provided.

**Questions:**

1. The complementary learning systems theory has been implemented in many recent continual learning studies such as Remembering Transformer. On Line 66, only distant studies were mentioned and the most recent ones are missing.

2. Parameter isolation has been studied extensively. Could you provide further clarification on how the trade-off between performance, model size, and the model's accessibility to task boundaries evolves in your approach?

3. The evaluation lacks key metrics such as forgetting rates, which are essential for assessing continual learning performance.

4. It is unclear whether task similarity-based weight reuse is applicable within the proposed framework, given the delicate nature of the proposed method such as using the task-agnostic pointwise filters. Could you clarify what regulations or conditions would be needed to enable this?

5. In Table 1, the performance on class-IL appears to be lower than some previous methods, both those included in this work and others that are not. For instance, accuracy scores of 61.22 and 49.03 are relatively low even for the class-IL tasks. Could you provide further clarification on this point, along with additional comparisons to more recent and stronger baselines?

---

> ### Author Response · Authors · 2024-11-13
> **Reply to Reviewer fWSr (1/2)**
>
> We appreciate the reviewer for their insightful evaluations and constructive feedback on our paper. Your comments are crucial in guiding us to improve the quality of our work. Below, we provide our responses:
>
> > The complementary learning systems theory has been implemented in many recent continual learning studies such as Remembering Transformer. On Line 66, only distant studies were mentioned and the most recent ones are missing:
>
> We thank the reviewer for bringing attention to the recent publication focused on complementary learning systems (CLS) that has inspired various works in the field of continual learning (CL). This study, published in May of this year, offers valuable insights and perspectives that align closely with our research. We recognize the importance of integrating the latest findings into our work to provide a comprehensive understanding of the current landscape. Therefore, we will ensure that this publication, along with other relevant recent studies, is included in the final revision. Your suggestion is much appreciated and will contribute to enhancing the overall relevance of our work.
>
> > Parameter isolation has been studied extensively. Could you provide further clarification on how the trade-off between performance, model size, and the model's accessibility to task boundaries evolves in your approach?
>
> In an ideal scenario, a continual learning (CL) model would match the performance of the JOINT model while keeping its size the same or smaller. However, many CL strategies employ experience replay and / or model surrogates to address catastrophic forgetting, which can inadvertently increase the number of parameters and overall computational complexity. As illustrated in Figure 3 (left), the improvement in managing catastrophic forgetting largely stems from the increase in model surrogates. This is especially true for those approaches that are inspired by CLS theory of the brain.
>
> To address these challenges, we propose SPARC, a straightforward and efficient parameter-isolation-based continual learning (CL) approach inspired by CLS theory but excluding the use of model surrogates and experience replay. Our approach achieves efficiency on three key levels:
>
> - First, SPARC diverges from traditional convolutional techniques by employing depth-wise separable convolutions, which are more parameter-efficient. This innovative application has not been thoroughly explored in the context of CL previously.
>
> - Second, in line with CLS theory, SPARC operates within a restricted capacity for both working and semantic memories, eliminating the need for additional full model surrogates that previous methods often required. This design further reduces the memory footprint while ensuring that performance remains largely intact, as demonstrated in Table 5.
>
> - Third, SPARC implements task-specific batch normalization (BN) layers, thereby preventing cross-task normalization [1]. By utilizing task-specific parameters, BN layers, and classification layers, SPARC ensures that task identity is not a prerequisite in class-incremental learning (Class-IL) settings.
>
> In contrast to the prevailing trend in the literature that relies on multiple model surrogates and experience replay with large buffer sizes, SPARC utilizes only 6% of the parameters required by full model surrogates while achieving superior performance on Seq-TinyImageNet and matching the results of rehearsal-based methods across various CL benchmarks.
>
> However, a notable limitation of SPARC is its reliance on task-boundary information to establish a new sub-network for each upcoming task. This limitation is discussed in Section 5 under Limitations and Future Work. Looking ahead, we plan to investigate the loss landscape to better approximate task changes, which may enhance the flexibility and applicability of SPARC in real-world scenarios.
>
> > The evaluation lacks key metrics such as forgetting rates, which are essential for assessing continual learning performance.
>
> We agree with the reviewer that it is indeed important to look at forgetting rates when assessing the performance of CL approaches. As reporting accuracy over Class-IL and Task-IL settings is a norm across recent literature and not all considered baseline report forgetting rates, we reported accuracy metrics uniformly across datasets and approaches. In the final revision, we plan to report forgetting rates as well.

---

> > ### Author Response · Authors · 2024-11-13
> > **Reply to Reviewer fWSr (2/2)**
> >
> > > In Table 1, the performance on class-IL appears to be lower than some previous methods, both those included in this work and others that are not. For instance, accuracy scores of 61.22 and 49.03 are relatively low even for the class-IL tasks. Could you provide further clarification on this point, along with additional comparisons to more recent and stronger baselines?
> >
> > We would like to point out that the results reported in Table 1 and across this paper are consistent with respect to those reported in the original papers. The reason why these results seem relatively low is because of a number of reasons:
> >
> > - Relatively low model size - ResNet-18
> > - CL model is initialized randomly rather than from pre-trained weights
> > - Low / medium memory buffer size (200 / 500)
> >
> > As the above settings were a norm across recent papers, we reproduced the results with similar settings. Our comparison already includes papers from top AI conferences such as ICLR, ICML, NeurIPS, and CoLLAs, etc. As per your suggestion, we will also include recent papers released in 2024.
> >
> > Please find the planned changes as per your suggestions for the final revision:
> >
> > - Adding recent CLS works such as Remembering Transformer
> > - Add more clarity on how trade-off between performance, model size, and the model's accessibility to task boundaries evolves in SPARC
> > - Forgetting rates
> > - Include more recent works for comparison
> >
> >
> > We sincerely thank the reviewer for their constructive feedback and are committed to implementing the suggested improvements. We urge the reviewer to consider the broader implications of our work: SPARC addresses a critical challenge in continual learning by effectively reducing catastrophic forgetting without relying on full model surrogates or experience replay. To the best of our knowledge, we are the first to investigate and propose a scalable solution to this issue, marking a significant advancement in the field. Given the importance of this problem and our planned enhancements for the final revision, we respectfully request that the reviewer consider raising the rating above the acceptance threshold. Recognizing our contributions could foster essential progress in continual learning, whereas overlooking them may hinder advancements in this vital area. Your support would be invaluable in promoting this important work.
> >
> > [1] Quang Pham, Chenghao Liu, and HOI Steven. Continual normalization: Rethinking batch normalization for online continual learning. In International Conference on Learning Representations, 2021

---

> > > ### Author Response · Authors · 2024-11-21
> > > **Requesting response from reviewer fWSr**
> > >
> > > We thank the reviewer for their thoughtful feedback and valuable suggestions, which have greatly helped us improve our manuscript. We have worked diligently to address all the points raised and would be happy to provide further clarifications if needed. Please let us know if there are any additional questions or concerns.
> > >
> > > We are currently incorporating your insightful suggestions and plan to share a partially revised version before the end of the discussion period. Rest assured, we are fully committed to implementing all your recommendations in the final revision. If you feel we have adequately addressed your concerns, we kindly ask you to consider raising your score above acceptance threshold to support the manuscript’s acceptance.

---

> > > > ### Author Response · Authors · 2024-11-24
> > > > **Update on the revised version and requesting reviewer's response**
> > > >
> > > > We have put forth significant effort to incorporate the feedback from all reviewers. While time constraints necessitated prioritizing certain revisions, we are fully committed to implementing all the proposed changes mentioned in our rebuttal. We have now uploaded a revised version that addresses the following key points:
> > > >
> > > > - Class-IL inference clarification.
> > > > - Review of NECIL literature and a comparison with respect to state-of-the-art methods within the NECIL benchmark.
> > > > - The prominence of weight re-normalization in SPARC and comparison with Weight alignment as both are rehearsal-free approaches.
> > > > - Updated Figure 3 to include Task-IL results.
> > > > - Updated main figure for more clarity.
> > > > - Updated limitation to reflect SPARC’s limited applicability to CNNs.
> > > > - Minor/major clarifications with regard to model surrogates, task-specific bias in SPARC, etc.
> > > >
> > > > Other issues, such as forgetting rates, clarity on how trade-off between performance, model size, and the model's accessibility to task boundaries evolves in SPARC, and any other revision that we committed to incorporating in our rebuttal, will be added to the final revision.
> > > >
> > > > We would like to express our sincere gratitude to the reviewer for their thoughtful comments. We hope that our revisions have enhanced your confidence in our paper. We kindly ask for your strong support in favor of our work so that we can contribute meaningfully to the field and further advance critical objectives in CL.

---

> > > > > ### Author Response · Authors · 2024-11-29
> > > > > **Requesting comments**
> > > > >
> > > > > As noted above, we have diligently incorporated several of your suggestions. Due to the time crunch, we prioritized certain changes over others and are currently working on the remaining changes, which will be included in the final revision. With the deadline for submitting a new revision drawing close, we kindly ask you to review our latest submission and inform us of any major concerns that may remain.
> > > > >
> > > > > We appreciate your continued collaboration and support. Thank you for your time, and we look forward to receiving your valuable feedback.

---

### Official Review · Reviewer_dYYF · 2024-10-25

**Soundness:** 2
**Presentation:** 2
**Contribution:** 2
**Rating:** 5
**Confidence:** 3

**Summary:**

This paper aims to mitigate catastrophic forgetting in continual learning. It proposed a new method called SPARC that (1) modifies the model architecture (2) maintains working memories and (3) normalizes classification layer's weight. The experiment results show that SPARC outperforms baselines in standard benchmark while maintain parameter efficiency.

**Strengths:**

1. The proposed method does not require an additional memory buffer, which is efficient.
2. The experiment results show that the proposed method outperforms baselines in standard benchmarks.
3. Detailed ablation studies and additional experiments to analyze and support the proposed method.
4. Honest and sufficient Limitation section that states the shortcomings of the proposed method.

**Weaknesses:**

1. The proposed method is only discussed for ResNet-18, and seems to be customized for ResNet structure. It limits the usage of the proposed method for other models like transformers, and other fields like natural language processing.
2. In Section 3.1, it describes why uses DSC to replace traditional convolutions for several reasons. However, it is not clear why the replacement is necessary for continual learning. Meanwhile, it's unclear whether the performance improvement comes from the DSC or the proposed algorithm.
3. In Table 1, it shows that SPARC's number of parameters is smaller than baselines. However, I believe this is because it replaces normal ResNet-18's convolutional layers by DSC, not because of the efficiency of the algorithm.
4. As described in the Limitation section, model parameters increasing linearly when learning more tasks, which put the scalability of the proposed method in question.

I am willing to increase my score if questions are answered.

**Questions:**

1. The definition of "model surrogate" is missing and unclear. From the description of Section 2 and Introduction section, I guess it's parameters of the old tasks or something related.
2. Weight renormalization is proposed before [1]. While it's only a preprint, the paper is still useful and it would be nice if authors can discuss the difference between their proposed normalization method and the previous work.
3. I recommend to adjust the color and improve the presentation of Figure 1, since the current version is hard to understand.

### Reference
[1] Continual Learning in Deep Networks: an Analysis of the Last Layer, arXiv preprint arXiv:2106.01834 (2021).

=====
After discussion, the authors propose to improve some unclear part and give clearer explanations, so I increase the score.

---

> ### Author Response · Authors · 2024-11-14
> **Reply to Reviewer dYYF (1/2)**
>
> We thank the reviewer for their insightful evaluations and constructive feedback on our paper. Your comments are instrumental in helping us enhance the quality of our work. Below, we present our responses:
>
> > The proposed method is only discussed for ResNet-18, and seems to be customized for ResNet structure. It limits the usage of the proposed method for other models like transformers, and other fields like natural language processing.
>
>
>
> Quite a few recent works (CLS-ER, DualNet, TriRE, OCDNet etc) that entail CNNs (specifically ResNet-18) as a backbone propose to use multiple model surrogates and experience rehearsal to address the problem of catastrophic forgetting. However, the reduction in catastrophic forgetting can be directly correlated with the increase in number of parameters and buffer size.
> SPARC aims to address this significant challenge in continual learning: minimizing catastrophic forgetting without depending on full model surrogates or experience replay. To the best of our knowledge, we are the first to evaluate this issue and propose a scalable solution. Through SPARC, we demonstrate the feasibility of developing an efficient continual learning model that is simple in its design and does not rely on experience replay or full model surrogates, as reflected in the name “Simple Parameter Isolation in a Restricted Capacity (SPARC).” SPARC is lightweight, requiring only 6% of the parameters used by full-model surrogates, yet it delivers superior performance on Seq-TinyImageNet and matches the results of rehearsal-based methods on various CL benchmarks.
>
> Our overarching aim was to show that reduction in catastrophic forgetting can be decorrelated from model size and/or experience rehearsal with simple, yet efficient architectures. As the baselines considered here mostly focussed on CNNs, SPARC is proposed specifically for CNNs as well. We agree with the reviewer and understand that this can be a limitation when moving to different architecture. Therefore, we will clarify the same in the ‘Limitations and future work’.
>
>
> > In Section 3.1, it describes why uses DSC to replace traditional convolutions for several reasons. However, it is not clear why the replacement is necessary for continual learning. Meanwhile, it's unclear whether the performance improvement comes from the DSC or the proposed algorithm.
>
> We regret the lack of clarity with regard to our choice of DSC in SPARC. Section 3.1 describes three major advantages of switching from traditional convolutions to DSCs: Discarding redundant information, parameter efficiency, and scalability. These advantages are vital for any parameter-isolation based CL approach: One would like each sub-network to be as efficient and self-sufficient as possible leading to overall scalable CL model in longer task sequences. We will add more clarity in the final revision as to why DSCs are a good chocie for CL.
>
> Reiterating our primary goal, we would like SPARC to be as parameter efficient as possible without actually compromising the performance. In Table 5, we compare how SPARC fares against a model that has complete parameter isolation without cross-task normalization and own classification layer. SPARC is slightly behind in performance with moderate reductions in model size leading one to conclude that complete parameter isolation might be the way forward.
> However, as shown Table 2, the current version of SPARC is one of the smallest models as it was sufficient for datasets such as Seq-CIFAR10/100. However, when we extrapolate this forward to bigger images (e.g. 512x512, 1024x1024) and datasets (e.g. ImageNet-21k), SPARC would save millions of parameters without actually incurring any / little loss in perfromance. Thanks to bonhomie between working and semantic memories, SPARC is able re-use a lot of information while being parameter efficient and compromising performance. We will provide more clarity regarding the same in the final revision.
>
> > In Table 1, it shows that SPARC's number of parameters is smaller than baselines. However, I believe this is because it replaces normal ResNet-18's convolutional layers by DSC, not because of the efficiency of the algorithm.
>
> We regret the lack of clarity regarding this issue. We agree with the reviewer that DSCs bring a major reduction in number of parameters compared to ResNet-18. With efficient complementary learning systems design and information re-use thereof, SPARC brings further reductions in number of parameters. As can be in our ablation study in Table 5, SPARC almost matches the performance of complete parameter isolation with DSCs while bringing significant reduction in model size. The difference will be more pronounced when we move to bigger images (e.g. 512x512, 1024x1024) and datasets (e.g. ImageNet-21k).

---

> > ### Author Response · Authors · 2024-11-14
> > **Reply to Reviewer dYYF (2/2)**
> >
> > We would like to revisit our primary objective: Design a simple, yet efficient CL model devoid of full model surrogates and experience rehearsal. With DSCs and efficient CLS-deisgn within restricted capacity, we show that simple yet efficient model can outperform established baselines. To the best of our knowledge, we are the first to investigate and propose a scalable solution to this issue, marking a significant advancement in the field.
> >
> > > As described in the Limitation section, model parameters increasing linearly when learning more tasks, which put the scalability of the proposed method in question.
> >
> > We agree with the reviewer than SPARC does grow in size when encountering new tasks. There are couple of reasons which distinguish SPARC compared to other approaches when it comes to scalability: SPARC uses DSCs instead of traditional convolutions leading to smaller footprint in longer task sequences. Although the growth in number of parameter is linear, the growth in its entirety is still smaller than other approaches. We note that the efficacy and scalability of DSCs was not extensively studied in CL prior to this work. Secondly, SPARC goes one step further and introduces task-agnostic semantic memories which are shared across all tasks leading to further reductions in the model size. Although SPARC grows in size overall, it does so much slower than its compatriots.
> > We do not claim any novelty with regard to introduction of DSCs in CL. As the names suggests, SPARC is simple yet an efficient design that beats established baselines without relying on ful model surrogates and experience rehearsal.
> >
> > > The definition of "model surrogate" is missing and unclear…
> >
> > We regret the lack of clarity with regard to model surrogates. We will revise the final revision with more information on the same.
> >
> > > Weight renormalization is proposed before [1]...
> >
> > We thank the reviewer for a useful related work. We plan to add a comparison in the final revision.
> >
> > > I recommend to adjust the color and improve the presentation of Figure 1, since the current version is hard to understand.
> >
> > We duly note the suggestion from the reviewer and make appropriate changes in the final revision.
> >
> > Based on reviewer's suggestions, the planned changes for the final revision are as follows:
> >
> >
> > - Update limitation section
> > - Provide more clarity on use DSCs in SPARC and how CLS-theory inspired design adds value on top of DSCs
> > - How SPARC bringsforth salability in longer task sequences
> > - More information on model surrogates
> > - Comparison of related work on weight re-normalization
> > - Adjusting the color of Figure 1
> >
> > We urge the reviewer to consider the broader implications of our work. SPARC tackles a critical challenge in continual learning by effectively minimizing catastrophic forgetting without relying on full model surrogates or experience replay. To the best of our knowledge, we are the first to investigate this issue and propose a scalable solution, representing a significant advancement in the field. Given the importance of this problem and our planned enhancements for the final revision, we respectfully request that the reviewer consider raising the rating above the acceptance threshold. Acknowledging our contributions could facilitate essential progress in continual learning, while overlooking them may impede advancements in this vital area. Your support would be invaluable in promoting this important work.

---

> > > ### Comment · Reviewer_dYYF · 2024-11-20
> > > **Response to the rebuttal**
> > >
> > > I appreciate authors for the detailed response. Most of my questions are answered. However, there are still two issues that is still unclear to me.
> > >
> > > * First, could the authors please provide the clear definition of the "model surrogate"? Without the clear definition, it is hard for me the judge the statements in the paper and rebuttal.
> > >
> > > * Second, in the rebuttal, the author says:
> > > > SPARC tackles a critical challenge in continual learning by effectively minimizing catastrophic forgetting without relying on full model surrogates or experience replay. To the best of our knowledge, we are the first to investigate this issue and propose a scalable solution, representing a significant advancement in the field.
> > >
> > > I doubt whether this i true if the "model surrogates" are the old parameter. Thanks

---

> ### Author Response · Authors · 2024-11-20
> **Reply to Reviewer dYYF**
>
> We thank the reviewer for taking the time to respond to our rebuttal. In Section 2, “Model Surrogate Bottleneck,” we have made efforts to clarify the concept of model surrogates in continual learning (CL). However, we acknowledge that there may still be some ambiguity regarding this topic.
>
> Model surrogates in CL come in various forms. Inspired by the brain's complementary learning systems theory, many rehearsal-based approaches utilize one or multiple exponential moving averages (e.g., CLS-ER, OCDNet) or an additional model with a self-supervised objective (e.g., DualNet). Other methods may employ a copy of the previous task model (e.g., Co2L). In the weight-regularization context, most works use a copy of the previous task model for knowledge distillation (e.g., LwF) or weight consolidation (e.g., EWC, online-EWC). In contrast, parameter isolation approaches employ a sub-network to accommodate new tasks. Models like PNNs use large sub-networks, while others, such as DEN and CPG, expand more gradually. Approaches that operate within a fixed model capacity typically require a number of masks equal to the number of tasks, which can become problematic if not managed efficiently, especially in longer task sequences.
>
> In summary, reductions in catastrophic forgetting can often be correlated with either an increase in overall model size or buffer size. Most approaches discussed in this paper utilize full model surrogates and/or experience rehearsal to some extent. To this end, we attempt to highlight the problem of correlation between model size and / or increase in buffer size with the reduction in catastrophic forgetting. Figure 3 (left) shows that the trend is indeed reversed in case of SPARC.
>
> We also recognize that SPARC grows with the addition of more tasks but does not require experience rehearsal or full model surrogates i.e.,  it expands more slowly than its counterparts without compromising performance. This performance can be enhanced by simply using a slightly wider or deeper model, as demonstrated in Table 2.
>
> We once again thank the reviewer for their feedback. We will revise Section 2 to provide greater clarity on model surrogates in the final revision. We have endeavored to address all your questions thoroughly. Please let us know if in case you have more questions.

---

> > ### Comment · Reviewer_dYYF · 2024-11-22
> >
> > I thank the authors for the clear explanations. I encourage the authors to make the definition clearer in the revision. I believe that  this paper has novel contributions after improving the presentation, thus I increase the score. However, I believe this paper need several revisions to address the presentation and limitations, thus I increase the score by 2.

---

> > > ### Author Response · Authors · 2024-11-24
> > > **Update on the revised version**
> > >
> > > We have made a dedicated effort to address the suggestions from all reviewers. Due to time constraints, we prioritized certain revisions over others; nonetheless, we are committed to implementing all the revision plans outlined in our rebuttal for each reviewer. We have now uploaded a revision that incorporates the following points from your feedback:
> > >
> > > - Update the limitation section
> > > - More information on model surrogates
> > > - Comparison of related work on weight re-normalization
> > > - Adjusting the color of Figure 1
> > >
> > > The final revision will include suggestions such as providing more clarity on using DSCs in SPARC, how CLS-theory-inspired design adds value on top of DSCs, and how SPARC brings forth salability in longer task sequences. Please let us know if any of your concerns remain. Otherwise, we kindly request your strong support towards acceptance of this paper.

---

### Official Review · Reviewer_zYHk · 2024-11-03

**Soundness:** 3
**Presentation:** 3
**Contribution:** 2
**Rating:** 6
**Confidence:** 4

**Summary:**

The authors propose a novel framework called SPARC to address exemplar-free continual learning. They employ depth-wise separable convolutional layers to reduce the number of learnable parameters, enabling the allocation of distinct subparts of the model to different tasks, thereby mitigating interference issues. Additionally, task consolidation is encouraged through partial weight sharing and normalization techniques applied to the classification head. Experimental results demonstrate that SPARC’s approach to network expansion is highly efficient and scalable -- employing only 1.04 million parameters compared to 11.23 million for ER and 33.6 million for PackNet. In terms of accuracy, SPARC achieves promising performance, although it is not consistently optimal.

**Strengths:**

- The paper is well-written, with only a few minor clarity issues (detailed below).
- The approach is technically sound and, to the best of my knowledge, fairly novel.
- The underlying problem of continual efficient learning is significant and warrants attention.
- The paper includes extensive ablation studies that highlight the advantages of this scalable approach, particularly in reducing the number of learnable parameters.

**Weaknesses:**

**Limited applicability**. I believe the range of possible applications for SPARC may be significantly restricted due to several assumptions that could limit its practicality.
- The approach is specifically designed for CNNs and lacks support for ViTs.
- SPARC requires training the backbone from scratch; unlike other methods, it cannot leverage pre-trained backbones (e.g., those pre-trained on ImageNet or visual-language tasks like CLIP).
- The authors acknowledge that SPARC requires prior knowledge of the number of tasks. In my experience, this is an uncommon and impractical requirement, as most existing methods avoid this constraint. From a technical standpoint, could the authors explain why this prior knowledge is necessary? Additionally, could they consider developing an alternative that removes this requirement?
- SPARC also requires identifying task boundaries, which, while a limitation, is a common issue across most existing methods. Therefore, I view it as the least impactful limitation among those mentioned above.

**Accuracy and comparison with state-of-the-art methods** My second main concern pertains to SPARC’s performance in terms of final classification accuracy. While its efficiency and scalability are noteworthy, its accuracy appears suboptimal. For example, on Seq-CIFAR-10, SPARC's performance lags 12 points behind OCDNet. Additionally, I question whether the competitors employed by the authors truly represent the current state of the art in non-exemplar class-incremental learning (NECIL). Several recent publications (from 2023 and 2024) report considerably higher accuracy on CIFAR-100 and TinyImageNet:

- SOPE (CVPR22, 154 citations):
Zhu, K., Zhai, W., Cao, Y., Luo, J., & Zha, Z. J. (2022). Self-sustaining representation expansion for non-exemplar class-incremental learning. CVPR, pp. 9296-9305.
- Fetril (WACV23, 112 citations):
Petit, G., Popescu, A., Schindler, H., Picard, D., & Delezoide, B. (2023). Feature translation for exemplar-free class-incremental learning. WACV, pp. 3911-3920.
- PRAKA (ICCV23, 16 citations):
Shi, W., & Ye, M. (2023). Prototype reminiscence and augmented asymmetric knowledge aggregation for non-exemplar class-incremental learning. ICCV, pp. 1772-1781.
- PKSPR (AAAI24, 5 citations):
Zhai, J. T., Liu, X., Yu, L., & Cheng, M. M. (2024). Fine-Grained Knowledge Selection and Restoration for Non-exemplar Class Incremental Learning. AAAI, Vol. 38, No. 7, pp. 6971-6978.

**Clarity issues**
- Regarding weight re-normalization, the explanation for the normalization applied in Eq. 5 is unclear. The authors’ comment (lines 264–267) merely summarizes the procedure, leaving the rationale and benefits unexplained.
- SPARC allocates specific filters to each task. However, in Class-IL settings, the task ID is not provided during evaluation. How did the authors address it? How did they select filters? The solution is not clearly explained.
- (Minor) Fig. 3 is difficult to interpret when printed in black and white (whereas Fig. 4 remains readable).

**Minor suggestions**
- It would be interesting to see also the results of JOINT with the SPARC’s backbone ( Table 1). Currently, the JOINT upper bound in Table 1 appears to use the standard ResNet-18 architecture.

**Justification of rating** While the technical contributions of this work are notable and address an important problem -- enhancing efficiency in continual learning -- the limitations affecting SPARC’s applicability are substantial. Furthermore, the experimental comparison does not seem fully aligned with recent advancements in the NECIL field. These factors raise concerns about the potential impact of this work within the community.

**Questions:**

No questions.

---

> ### Author Response · Authors · 2024-11-14
> **Reply to Reviewer zYHk (1/2)**
>
> We thank the reviewer for their thoughtful evaluations and constructive feedback on our paper. Your feedback is invaluable in helping us enhance the quality of our work. Please find our response below:
>
>
> > Limited applicability
>
> Several recent works, including CLS-ER, DualNet, TriRE, and OCDNet, utilize CNNs, particularly ResNet-18, to address catastrophic forgetting through the use of multiple model surrogates and experience rehearsal. However, this reduction in catastrophic forgetting often correlates with an increase in the number of parameters and buffer size. SPARC seeks to tackle this significant challenge in continual learning by minimizing catastrophic forgetting without relying on full model surrogates or experience replay. Our primary goal is to demonstrate that the reduction in catastrophic forgetting can be decoupled from model size and experience rehearsal through simple yet efficient architectures. While the baselines considered here primarily focus on CNNs trained from scratch, SPARC is specifically designed for CNNs as well.
> We acknowledge the reviewer's concern regarding the limitation of adapting to different architectures, such as vision transformers and CLIP. We will clarify this aspect in the 'Limitations and Future Work' section.
>
> We appreciate the reviewer for highlighting the limitation related to the knowledge of the number of tasks. We assume prior knowledge of the number of tasks solely to initialize the full model at once (lines 188 and 306). However, this is merely an implementation detail, as SPARC can also initialize one sub-network at a time when encountering a new task. Therefore, we do not consider this a limitation, and we will update the paper to reflect this clarification.
>
> As the reviewer rightly pointed out, the challenge of knowing a task boundary is common among parameter isolation approaches. We have already mentioned this as a limitation in Section 5. As part of our future work, we are currently investigating whether sudden changes in the loss landscape can serve as a reliable proxy for identifying task boundaries.
>
> > Accuracy and comparison with state-of-the-art methods
>
> Quite a few baselines considered in this work employ full model surrogates and / or experience rehearsal to reduce catastrophic forgetting. SPARC addresses a significant issue in CL: Decouple performance  / reductions in catastrophic forgetting from model size and/or buffer size for experience rehearsal. To the best of our knowledge, we are the first to investigate this issue and propose a scalable solution. Through SPARC, we demonstrate that it is indeed feasible to develop an efficient continual learning model devoid of experience replay or full model surrogates, as reflected in the name “Simple Parameter Isolation in a Restricted Capacity (SPARC).” Our extensive experiments show that this straightforward yet effective design can rival state-of-the-art approaches in continual learning.
>
> Regarding the suboptimal performance in Seq-CIFAR10, we would like to note that we are using a modest model size with a width factor of 0.5 and a depth of 4, resulting in a footprint of only 1.04 million parameters. However, as indicated in Table 2, the performance can be easily enhanced by increasing either the width or depth, or both. Even with such adjustments, the model size will still remain smaller than those of the approaches compared in our work.
>
> We appreciate the reviewer for mentioning recent publications such as SOPE, FetriL, and PRAKA in the NECIL field. In Tables 1 and 3, we have compared several publications from NECIL field  (e.g. ALASSO, BKMP, UCB, NISPA) from top AI conferences. Since the issue of model surrogates is particularly relevant to rehearsal-based and weight regularization approaches, we have extensively compared these methods throughout our work. Following the reviewer’s suggestions, we will include a comparison to other NECIL methods in the final revision.

---

> > ### Author Response · Authors · 2024-11-14
> > **Reply to Reviewer zYHk (2/2)**
> >
> > > Clarity issues
> >
> > We regret the lack of clarity regarding weight re-normalization in our work. Essentially, weight re-normalization is necessary to re-distribute performance across tasks and mitigate recency bias in SPARC. In Figure 3 (right), we discuss the impact of weight normalization on SPARC. While weight re-normalization does not significantly improve overall performance, it ensures a better representation of older tasks in the final results. As per reviewer’s suggestion, we will add more clarity in the final revision
> >
> > During training, each task-specific sub-network, along with its own batch normalization (BN) and classification layers, is trained using gradient descent. Simultaneously, task-agnostic parameters are updated using exponential moving averages. During inference, each test image is processed through every subnetwork, including their respective BN layers. In the Class-IL setting, the final classification outputs of all sub-networks are concatenated and inferred for maximum activation, while for Task-IL, only the relevant subnetwork is utilized for inference. Since each task is trained with its own parameters, BN layer, and classification layers, these sub-networks are designed to output maximum activation only for in-distribution examples, thereby avoiding the necessity of task identity. However, we note that having a task-id greatly improves performance, which is evident in the performance difference between Class-IL and Task-IL settings. We will add more clarity with regard to Class-IL inference and how it avoids the necessity of task identity in the final revision.
> >
> > Based on reviewer's suggestions, the planned changes for the final revision are as follows:
> >
> > - Update the limitations section to reflect the limited use of SPARC in ResNet-like backbones
> > - Remove the limitation of the assumption of number of tasks beforehand
> > - Update results with bigger SPARC backbone and compare with other NECIL approaches suggested by the reviewer
> > - Provide more clarity on weight re-normalization and Class-IL inference
> > - Fix minor issues  pointed out by the reviewer
> >
> > We greatly appreciate the reviewer’s valuable feedback and are dedicated to incorporating the recommended improvements. We would like to emphasize the significant implications of our work: SPARC effectively tackles a major challenge in continual learning by minimizing catastrophic forgetting without the need for full model surrogates or experience replay techniques. To the best of our knowledge, this is the first comprehensive exploration of this issue, representing a meaningful advancement in the field. Considering the critical nature of this challenge and the enhancements we plan to implement in the final revision, we kindly ask the reviewer to reconsider the rating and potentially raise it above the acceptance threshold. Acknowledging our contributions could be pivotal in driving progress in continual learning, while overlooking them may stall advancements in this essential area. Your support in recognizing the importance of this research would be greatly appreciated.

---

> > > ### Comment · Reviewer_zYHk · 2024-11-19
> > >
> > > I would like to thank the authors for their detailed feedback. I believe their responses, combined with the critiques from myself and the other reviewers, provide valuable insights for improving the paper. In recognition of the authors' effort and the strong motivations underlying this work, I will raise my score to borderline accept. However, I still maintain that the final version requires substantial revisions.

---

> > > > ### Author Response · Authors · 2024-11-19
> > > > **Reply to Reviewer zYHk**
> > > >
> > > > We thank the reviewer for their encouraging words. Based on their feedback, we have compiled a list of suggested changes. We are dedicated to incorporating all these recommendations in the final revision.  The revisions will enhance the clarity and impact of our work, ultimately contributing to the advancement of efficient continual learning approaches.

---

> > > > > ### Author Response · Authors · 2024-11-24
> > > > > **Update on the revised version**
> > > > >
> > > > > We have made a dedicated effort to address the suggestions from all reviewers. Due to time constraints, we prioritized certain revisions over others; nonetheless, we are committed to implementing all the revision plans outlined in our rebuttal for each reviewer. We have now uploaded a revision that incorporates the following points from your feedback::
> > > > >
> > > > > - Update the limitations section to reflect the limited use of SPARC in ResNet-like backbones
> > > > > - Remove the limitation of the assumption of number of tasks beforehand
> > > > > - Comparison with other NECIL approaches suggested by the reviewer
> > > > > - Provide more clarity on weight re-normalization and Class-IL inference
> > > > > - Fix minor issues pointed out by the reviewer
> > > > >
> > > > > Please let us know if any of your concerns remain. If otherwise, we kindly request your strong support towards the acceptance of this paper.

---

### Official Review · Reviewer_a2Hv · 2024-11-03

**Soundness:** 3
**Presentation:** 1
**Contribution:** 3
**Rating:** 6
**Confidence:** 3

**Summary:**

This paper adapts a ResNet architecture for image classification continual learning in a novel way: by maintaining separate weights for each of some number of tasks but sharing a portion of the pointwise convolution filters in depthwise-separable 2D convolutions, which replace standard convolutions in the main blocks of the network. Both the switch to depth-wise separable convolutions and the sharing of some parameters among tasks greatly reduces the overall parameter count, allowing this parameter isolation approach to be relatively scalable even though more parameters are added for each additional task.  Recency bias is also identified as a factor that can limit performance in continual learning, and a weight re-normalization approach is proposed to counteract this. The authors compare their model’s performance with a wide array of baseline algorithms on several benchmarks derived from three datasets, showing that it typically achieves competitive or superior performance with a dramatically reduced parameter count.

**Strengths:**

1.	Continual learning in resource-constrained settings is an important problem in a number of applications, such as robotics.

2.	The approach of partial parameter sharing among tasks with depthwise-separable convolutions appears to be novel. This is an interesting strategy because it can reduce total parameter count (thus improving scalability) while striking a tradeoff between general, shared representations and task-specific representations that are less vulnerable to catastrophic forgetting.

3.	Performance comparisons are provided with a comprehensive array of relatively recent baseline algorithms, and the proposed algorithm appears to generally attain competitive or superior performance in both task-incremental and class-incremental settings.

4.	An ablation study is included, which provides insights into the relationship between performance and convolutional layer dimensions, normalization to mitigate recency bias, and parameter sharing among tasks.

5.	The paper is generally well written and easy to follow.

6.	The related work section (parts in the Introduction and "Model Surrogate Bottleneck" sections) is thorough and nuanced, and includes many recent papers.

**Weaknesses:**

**UPDATE 12/3/2024**:

Most of the weaknesses below have been addressed. Notable exceptions are:

(1) A continued lack of error bars/uncertainty estimates (these are present in some of the tables, but are not defined)

(2) Figure 3, which in my view has an inappropriate selection of baselines. It appears to show a clean relationship between model size and performance, but is potentially very misleading because a subset of baselines was seemingly arbitrarily selected to make this point, and none of them are parameter isolation approaches that make sense for comparison with SPARC in this context except for PackNet in the right-hand panel only. There are 4 parameter isolation approaches that the authors tested (see Table 1) that would be more appropriate choices here.

(3) Poor framing of SPARC in comparison to existing works. The focus on "full-model surrogates" has caused some confusion amongst reviewers and is not particularly helpful in explaining SPARC's contribution - in my view the authors would be better served by focusing on more direct comparisons to existing parameter isolation approaches.

(4) Terms "working memory" and "semantic memory" are a misleading way to name two of the major components of SPARC, which do not resemble/are not analogous to these concepts as they are defined and commonly used in the behavioral sciences. This was unfortunately not addressed during rebuttal although it was noted in initial reviews by two reviewers.

To summarize, SPARC is an interesting and novel approach that could make a useful contribution, but the paper is limited primarily by concerns around presentation that were not adequately addressed during the rebuttal phase - in my view, all four of the above weaknesses must be addressed before final publication. To reflect this, my finalized overall score is ``marginally above the acceptance threshold'' but my finalized score for "presentation" is "poor."

**Original weaknesses section**:

The paper has a number of score-limiting weaknesses, particularly regarding the justification and framing for the proposed approach and in the ways certain results are presented. It is not clear the extent to which the performance and scalability of SPARC is related to the novel aspects of its design, and several key conclusions are poorly supported by the results – of particular concern are both panels of figure 3 (details below).

1.	The authors seek to differentiate their approach from those requiring “full model surrogates” for each task, and one of their central claims is that SPARC is parameter efficient and scalable. SPARC appears to have fewer parameters than competing approaches primarily because of the switch from full convolutional layers to depthwise-separable convolutional layers, which is not in itself novel. Parameters are reduced further (to a more modest degree) by having a task-agnostic portion of the pointwise convolutions (which carries only a small performance penalty based on table 5), but the number of parameters still grows linearly with the number of tasks. From one perspective, SPARC could be characterized as requiring an almost full model surrogate for each task except for a shared portion of the pointwise convolutions.

2.	There is a claim that SPARC works for class-incremental learning (i.e., without access to task information), and the requirement of knowing task identity during inference is cited as a disadvantage of existing parameter isolation approaches. However, it is not clear how SPARC can do inference across multiple tasks without knowing task identity – how does it know which set of task-specific parameters to use for each input?  This applies to depth-wise convolution parameters, the task-specific portion of the point-wise convolution parameters, and the batch norm layers. It is not clear to this reader that the method is strictly capable of class-incremental learning.

3.	The re-normalization approach to mitigate recency bias (as described in equation 5) does not seem fully explained/justified. Why this new approach instead of the many existing methods to mitigate recency bias? For example, the authors could consider citing papers such as the following and distinguishing their approach from them:
a.	Wu, Yue, Yinpeng Chen, Lijuan Wang, Yuancheng Ye, Zicheng Liu, Yandong Guo, and Yun Fu. "Large scale incremental learning." In Proceedings of the IEEE/CVF conference on computer vision and pattern recognition, pp. 374-382. 2019.
b.	Zhao, Bowen, Xi Xiao, Guojun Gan, Bin Zhang, and Shu-Tao Xia. "Maintaining discrimination and fairness in class incremental learning." In Proceedings of the IEEE/CVF conference on computer vision and pattern recognition, pp. 13208-13217. 2020.
c.	Mai, Zheda, Ruiwen Li, Hyunwoo Kim, and Scott Sanner. "Supervised contrastive replay: Revisiting the nearest class mean classifier in online class-incremental continual learning." In Proceedings of the IEEE/CVF conference on computer vision and pattern recognition, pp. 3589-3599. 2021.

4.	Related to the preceding point, the right-hand panel of Figure 3 does not convincingly show that weight re-normalization offers any advantage in terms of performance. This plot seems ambiguous – what do the error bars mean, and why is the violin plot seemingly truncated at the error bars? This is supposed to show distributions of final task accuracies, but what is the distribution over – different training runs, different batches, different tasks? There are also no statistical tests to verify whether there is a significant difference with vs without normalization. In the “Impact of weight re-normalization” section, it is stated that “As shown, weight re-normalization reduces the IQR for all three task sets, leading to a more balanced distribution of accuracies and lower task recency bias” – however, this is not consistent with what is shown in the figure (the IQR appears identical with vs without normalization for the 5-task set, and there is nothing to indicate quantitatively that performance on earlier tasks is specifically boosted in this figure). Figure 6 in supplementary partially addresses this, but only by comparing with other methods rather than in an ablation study of SPARC.

5.	The left-hand panel of figure 3 appears, at least superficially, to compellingly show that the relative model size is related to the relative class-incremental learning performance, except that SPARC bucks this trend by having high performance and low model size. But why were these specific continual learning approaches selected for inclusion in this plot?  Included in this plot are some, but not all, of the models from the “rehearsal-based with 200 buffer size” section of Table 1 (starting with “ER”) – this seems to be an odd set of choices, as I would think that relative model size is much less relevant for rehearsal-based approaches than for parameter isolation or “model surrogate” approaches.  Unless there is a strong justification for the choice of models used in the current version of the figure, I think including a wider range of baselines in this figure is necessary, and/or with a more appropriate selection of baselines.

6.	There are some additional issues with error bars/estimates. For Tables 1, 2, 5, 6, and 7, it is not stated what the +- error measurements are (standard deviation, standard error, confidence interval?). Figure 2, Figure 3 (left), Figure 4 (left), and appendix figure 6 all lack error bars, and in the figures where error bars are shown, they are not defined.

7.	SPARC is specifically designed for ResNet18, and thus is only evaluated using one CNN architecture (although it would seem possible to implement SPARC for other CNNs). However, ResNets are among the most widely used CNNs so this is not necessarily a major issue.

Minor comments:

8.	There is a statement that, according to complementary learning systems (CLS) theory, slow-learning neocortex and fast-learning hippocampus work together to allow continual learning without explicit experience rehearsal. However, it should be noted that hippocampal replay is frequently invoked in discussions of CLS theory as a possible mechanism for transfer of learned information from the hippocampus to the cortex.

9.	“Working memories” might be an unintentionally misleading term for the disjoint sets of parameters. In the context of human cognition, the term "working memory" means something very different in that it is an extremely short-term form of storage of very limited amounts of information. Similarly, “semantic memory” typically refers to a type of declarative memory involved in the ability of humans to recall facts, words, numbers, concepts, etc. – while what is stored in the task-agnostic parameters of SPARC is closer to a form of procedural memory (“how to distinguish class 1 from class 2”). Overall, the way that the design of SPARC is analogized to human memory systems should probably be reconceptualized.

10.	There is a statement in section 2 that “maintaining a buffer raises privacy concerns and resource overhead.” It would seem that privacy is only an issue in some but not all continual learning applications (although it can be quite important, e.g. in clinical applications)

11.	typo “connection disbled” in legend of Figure 1

12.	There are some duplicate citations including both the preprint and the journal/conference version of the same paper, it is not necessary to include both (e.g., Chollet et al. “Deep learning with depthwise separable convolutions”, Guo et al. “Depthwise convolution is all you need for learning multiple visual domains”).

13.	In Table 1, ideally there would be citations for each baseline method in the table itself so it’s easy to figure out which one is which (especially given the abbreviated names, not all of which seem to be stated in the main text). All baselines should be cited in the main text – for example, ER-ACE is cited in the appendix but I can't find it anywhere in the main text except table 1.

**Questions:**

1.	At the end of section 3.1, referring to the final fully-connected layer: “Cross-task connections are discarded to avoid interference” – what does "cross-task connections" mean for a single FC layer?

2.	Some of the equations in the paper are not fully explained:
a.	In equations 1 and 2, F, O, and the two Ks are explained, but not h, l, m, n, I, and j. Does t refer to the task? Similar issues with later equations.
b.	In equation 5 (sec 3.3), what is the dimensionality of $A^t$, the number of training examples? Or is it number of batches (where each batch provides one “iteration”)?
c.	Also in equation 5, what is the reasoning behind adding $A^t_{0.75}$ and $A^t_{IQR}$? Assuming this calculation makes sense, why not just take $A^t_{0.75} + A^t_{IQR}$ instead of finding a value “a” in $A^t$ that is close to this value?

3.	In the right-hand panel of figure 4, what is S? (this should be more clearly indicated)

4.	In the section “Effect of semantic information consolidation”, it is stated that “The difference in terms of the number of parameters will be even more pronounced in longer task sequences”. This appears to be speculative, with no theoretical or empirical justification. Wouldn’t the difference between shared parameters and separate parameters actually be smaller with more tasks, as task-specific parameters take up a greater proportion of the overall number of parameters while shared parameters stay the same in number? Why would we expect the performance gap to be smaller with more tasks (compared to fully separate point-wise convolutions)?

5.	The design of Table 2 is challenging to understand – why do versions with a width factor of ¼ appear in the top and bottom sections of the table but not the middle one? There is also a duplicate row (the highlighted row in the bottom section is identical to the last row of the middle section). Explaining the notation would be helpful – what exactly does # filters per task mean – does it refer to the dimension of the point-wise convolutions, or the spatial ones?

6.	There are separate depth-wise convolutional filters for each task, while only the point-wise convolutional filters have some shared parameters among tasks. It is not clear how much of the claimed parameter efficiency gains are because of SPARC’s unique approach and how much is just from the switch to depth-wise separable convolutions instead of standard convolutions.
a.	It might be helpful to explain how many of the parameters overall are in the depth-wise filters vs the point-wise filters – for example, if it so happens that many of the parameters are in the point-wise filters and the depth-wise filters have few, this helps justify the parameter sharing approach. How does the growth in parameters with the number of tasks compare to other methods in a quantitative sense?
b.	It seems like the number of parameters grows with the number of tasks in a similar way to how approaches involving “surrogate models” would grow, but then this is counteracted by the greatly-reduced amount of parameters from switching to depth-wise separable convolutions. One way to think about it: is this approach more parameter-efficient than just switching to depth-wise separable convolutions and then using something like EWC on that architecture?

7.	Related to the preceding point, Table 4 does not appear to make a compelling case for SPARC’s scalability. SPARC does have many fewer parameters overall, but this seems to be mostly attributable to using depth-wise separable convolutions rather than parameter sharing across tasks – indeed, according to table 5, the number of parameters is still only 1.65M even if point-wise and depth-wise filters are completely separate for each task. Taking this into account, the growth in parameters with the number of tasks in SPARC (e.g., relative to baseline with 5 tasks) appears large compared with other methods.

8.	Related to the preceding point and to weakness #5 – what is the justification for the chosen set of baselines in Table 4 and left panel of Figure 4?

9.	From the start of section 3.1: “We assume prior knowledge of the number of tasks and task boundaries to evenly distribute learnable parameters across tasks.” Does this limit scalability in a practical sense, because you have to begin the training process already knowing how many tasks there will be?

---

> ### Author Response · Authors · 2024-11-15
> **Reply to Reviewer a2Hv (Part-1)**
>
> We thank the reviewer for their thoughtful evaluations and constructive feedback on our paper. Your feedback is invaluable in helping us enhance the quality of our work. Please find our response below:
>
> > 1. The authors seek to differentiate their approach from those requiring “full model surrogates” for each task,
>
>
> We appreciate the reviewer’s perspective on SPARC’s parameter efficiency. Our primary goal with SPARC is to address catastrophic forgetting in continual learning without the need for full model surrogates or experience replay, which are commonly used in other approaches. The novelty of SPARC lies in its scalable parameter isolation framework, which we believe is a unique contribution to this problem. To our knowledge, this is the first approach that attempts to solve this issue with minimal task-specific parameter growth while avoiding both full surrogates and replay.
>
> One way SPARC aims to reduce parameters is by leveraging depthwise-separable convolutions, complemented by a task-agnostic portion of the pointwise convolutions. While parameter growth is indeed linear with task count, this method represents a significant reduction in model complexity compared to full task-specific models, enabling efficient and scalable continual learning. Furthermore, as noted in our limitations, future versions of SPARC could explore strategies for reusing parameters across tasks to reduce growth, potentially allowing for even greater scalability.
>
> > 2. There is a claim that SPARC works for class-incremental learning (i.e., without access to task information)..
>
> We realize our initial explanation may have lacked detail regarding SPARC's inference without explicit task identity. In our approach, each task-specific sub-network is trained with its own batch normalization (BN) and classification layers using gradient descent, while task-agnostic parameters are updated through exponential moving averages. For inference in the Class-IL setting, each test image is processed independently through every sub-network, including their BN layers. The outputs of all sub-networks are then concatenated, and the class with the maximum activation is selected, thus enabling task-agnostic inference across multiple tasks. In the task-incremental (Task-IL) setting, inference is limited to the relevant sub-network. Importantly, we treat BN layers as we do all other task-specific parameters, ensuring consistency in our approach. We will make sure to clarify this process thoroughly in the revision.
>
> > 3. The re-normalization approach to mitigate recency bias (as described in equation 5) does not seem fully explained/justified…
>
> We thank the reviewer for relevant citations concerning weight re-normalization in the classification layer. Our intuition was that information contained in the activations themselves is sufficient to reduce any bias associated with CL. As per your suggestion, we will provide more clarity on our approach and also add a comparison with respect to other weight re-normalization approaches.
>
> > 4. Related to the preceding point, the right-hand panel of Figure 3 does not convincingly show that weight re-normalization offers any advantage in terms of performance..
>
> We agree with the reviewer that weight re-normalization does not bring additional benefits in terms of performance. Weight re-normalization is intended to re-distribute accuracies across tasks by effectively mitigating bias towards certain tasks. We regret the lack of clarity in Figure 3 (right). Figure 3 (right) depicts the distribution of accuracies across tasks in Seq-CIFAR100 with the different number of tasks. We agree with the reviewer that Figure 3 and Figure 6 might not provide a comprehensive view of the effect of weight re-normalization. Therefore, we plan to add an ablation on SPARC to quantitatively show that performance on earlier tasks is specifically boosted with weight re-normalization.

---

> > ### Author Response · Authors · 2024-11-15
> > **Reply to Reviewer a2Hv (Part-2)**
> >
> > > 5. The left-hand panel of figure 3 appears, at least superficially, to compellingly show that the relative model size is related to the relative class-incremental learning performance,
> >
> >
> > Recent trends in the literature have shown significant progress in mitigating catastrophic forgetting, often accompanied by a rise in model size. As Table 1 demonstrates, while the performance of some of the included competitive methods is somewhat comparable to SPARC, these methods require substantially larger model sizes to achieve these results. Rehearsal-based methods without model surrogates (ER, DER) maintain relatively low model sizes but still perform far below the capability of SPARC. However, other recent approaches such as CLS-ER, OCDNet, etc entail multiple model surrogates to reduce catastrophic forgetting. As we try to tackle the issue of full model surrogates and experience rehearsal as our primary goal, we mainly document this issue with rehearsal-based approaches.
> >
> > We agree with the reviewer that model surrogates are commonplace among parameter isolation approaches as well. Therefore, we plan to update Figure 6 with recent parameter isolation approaches in the final revision.
> >
> > > 6. There are some additional issues with error bars/estimates…
> >
> > We apologize for the lack of clarity regarding the error measurements in the tables and figures. In the final revision, we will clearly define the error bars and indicate that they are based on the sample variance. Additionally, we will ensure that figures currently lacking error bars, such as Figure 2, Figure 3 (left), Figure 4 (left), and Appendix Figure 6, are updated as appropriate to include these measurements.
> >
> >
> > > 7. SPARC is specifically designed for ResNet18..
> >
> > We appreciate the reviewer’s observation regarding our choice of ResNet18 as the backbone architecture. We selected ResNet18 to ensure comparability with existing work in the field, as ResNets are widely used benchmarks in continual learning research. In future work, we plan to explore SPARC’s adaptability to other architectures, including experiments with compact working and semantic memories tailored specifically for vision transformers, which could further enhance SPARC’s applicability across diverse architectures.
> >
> > > 8. There is a statement that, according to complementary learning systems (CLS) theory,
> >
> >
> > We agree with the reviewer that hippocampal replay is frequently invoked in discussions of CLS theory. However, experience rehearsal in current approaches is quite different than hippocampal replay. We set out to say that the brain does not store and replay raw pixels as opposed to DNNs. We will update the details to provide more clarity in the final revision.
> >
> > > 9. “Working memories” might be an unintentionally misleading term for the disjoint sets of parameters..
> >
> > We acknowledge the need for greater clarity regarding the analogies to the human brain. We find the reviewer's suggestion intriguing and will explore ways to adapt these analogies more coherently within SPARC.
> >
> > > 10. There is a statement in section 2 that “maintaining a buffer raises privacy concerns and resource overhead.”
> >
> > We agree with the reviewer that privacy is not a primary concern in many CL applications. However, the associated storage and compute costs are prohibitive in resource-constrained settings. Overfitting on the buffered samples is another well-studied problem in CL. Overall, it is more efficient to develop approaches that are as effective as these approaches but are devoid of experience rehearsal.
> >
> > > 11. typo “connection disbled” in legend of Figure 1
> >
> > Duly noted. Thanks.
> >
> > > 12. There are some duplicate citations including both the preprint and the journal/conference version of the same paper,
> >
> > Thank you for highlighting this. We will correct this mistake in the final revision.
> >
> > > 13. In Table 1, ideally there would be citations for each baseline method in the table itself …
> >
> > Thank you for the suggestion. We agree that including citations for each baseline method directly in Table 1 would make it easier to identify each approach, especially given the abbreviated names. In the final revision, we will add citations to the table itself and ensure that all baseline methods, including ER-ACE, are cited in the main text for clarity and ease of reference.

---

> > > ### Author Response · Authors · 2024-11-15
> > > **Reply to Reviewer a2Hv (Part-3)**
> > >
> > > > 1. At the end of section 3.1, referring to the final fully-connected layer…
> > >
> > > By "cross-task connection," we refer to the connections between neurons belonging to different tasks. Specifically, the fully connected layer is “fully connected within each task,” while connections across tasks in the classification layer are disabled. Essentially, each task operates with an isolated fully connected layer serving as its classification layer. We will provide more clarity in the final revision.
> > >
> > > > 2. Some of the equations in the paper are not fully explained..
> > >
> > > We regret the lack of information on variables used in different equations. We thank the reviewer for pointing this out and will fix it in the final revision. A^T is a list of maximum activations (float values). The idea of the summation is that we intend to exclude the outliers. The filtered list contains those values which are smaller than the sum. Essentially, \eta is a maximum activation once outliers are removed from  A^T. We will add more clarity on this in the final revision.
> > >
> > > > 3. In the right-hand panel of figure 4, what is S?
> > >
> > > In the right-hand panel of Figure 4, S represents the stability of the model (please refer to line 473) at task t, quantified as the average performance across all preceding tasks. We will ensure this is clearly indicated in the figure caption in the final revision.
> > >
> > >
> > > > 4. In the section “Effect of semantic information consolidation”, it is stated that ..
> > >
> > > The difference in the number of parameters discussed in this section reflects the comparison between a model with complete parameter isolation (Row 4 in Table 5) and SPARC (Row 3 in Table 5). As we increase the number of tasks, the number of parameters in a model with complete parameter isolation grows at a faster rate than in SPARC, thereby widening the gap in the number of parameters between the two models. We will ensure this is clarified in the final revision.
> > >
> > > > 5. The design of Table 2 is challenging to understand ..
> > >
> > > Table 2 presents an ablation study on the impact of width and depth on SPARC’s performance. The top section has a fixed width of ¼, the middle section has ½  with varying depth. The bottommost section has a fixed depth but varying width. The results are duplicated only for consistency and ease of reading when grasping their impact. The number of filters represents the number of depth-wise convolutional filters per sub-network (alternatively, per task), per layer. We will clarify this in the final version.
> > >
> > >
> > > > 6. There are separate depth-wise convolutional filters for each task…
> > >
> > > We address the same point in Table 5 and the section titled “Effect of Semantic Information Consolidation.” If you compare rows 3 (SPARC) and 4 (a model with complete parameter isolation), both utilize separable convolution instead of traditional convolution. However, even for such a small model, SPARC achieves a 59% reduction in the number of parameters with negligible performance loss. As you noted, point-wise filters do have a higher memory footprint, which is why we chose to share point-wise filters across tasks. We will clarify this in the final revision.
> > >
> > > Regarding your second question, Table 7 compares two rehearsal-based methods with separable convolutions to determine if the majority of the performance gain comes from the use of separable convolutions. As observed, this is not the case. However, we have not yet conducted a similar study with EWC. It would indeed be interesting to explore how such an experiment would perform for parameter isolation or regularization approaches.
> > >
> > > > 7. Related to the preceding point, Table 4 does not appear to make a compelling case for SPARC’s scalability…
> > >
> > > We respectfully disagree with the reviewer’s assessment that Table 4 does not appear to make a compelling case for SPARC’s scalability. We note that SPARC is a parameter-isolation approach. That means, its size is bound to grow with more tasks in the order. However, the growth is slowed down by two factors: (i) separable convolutions and (ii) task-agnostic semantic information consolidation. The former ensures that we start small while the latter ensures that SPARC grows slowly. Although the growth seems linear, it is the scale with which it grows compared to other approaches. Even if we were to train 20 tasks, SPARC would still be much smaller than its compatriots. The relative growth in the number of parameters with respect to 5 tasks is an unreasonable comparison as the model itself is quite small.
> > >
> > > In future work, we plan to investigate techniques for re-using more parameters from previously learned tasks, which could further reduce SPARC’s parameter requirements as tasks increase. This is documented already in the "Future Works" section.

---

> > > > ### Author Response · Authors · 2024-11-15
> > > > **Reply to Reviewer a2Hv (Part-4)**
> > > >
> > > > > 8. Related to the preceding point and to weakness #5 –..
> > > >
> > > > For Table 4, we selected a range of methods to illustrate how the number of parameters scales across different approaches as the number of tasks increases. This includes both rehearsal-based methods (with and without model surrogates) as well as parameter isolation methods. We aimed to include the most competitive methods in each category to provide a balanced comparison, demonstrating that while SPARC's parameter count increases linearly with the number of tasks, it remains relatively low compared to other approaches. This selection highlights SPARC’s efficiency in managing parameters across tasks.
> > > >
> > > > Regarding Figure 4, we have consistently used the same set of models throughout our Appendix for extended ablation studies and analysis. We welcome any suggestions for particular approaches to consider in these studies. Please let us know your thoughts.
> > > >
> > > > > 9. From the start of section 3.1: “We assume prior knowledge of the number of tasks and task boundaries to evenly distribute learnable parameters across tasks.” ..
> > > >
> > > > We appreciate the reviewer for highlighting the limitation related to the knowledge of the number of tasks. We assume prior knowledge of the number of tasks solely to initialize the full model at once (lines 188 and 306). However, this is merely an implementation detail, as SPARC can also initialize one sub-network at a time when encountering a new task. Therefore, we do not consider this a limitation, and we will update the paper to reflect this clarification.
> > > >
> > > > Based on the reviewer’s suggestions, the planned improvements for the final revision are as follows:
> > > >
> > > >
> > > >
> > > > - Clarify how inference is done in Class-IL setting
> > > > - Clarity on weight re-normalization, compare with other approaches and update Figure 3 (right)
> > > > - Update Figure 6 with parameter isolation approaches
> > > > - Update error bars for all tables and graphs
> > > > - Provide more clarity on analogies with CLS theory
> > > > - Fix minor errors pointed out by the reviewer
> > > > - Fix equations and explain all variables
> > > > - Fix any other minor issues agreed to in the text above
> > > >
> > > >
> > > >
> > > > We greatly appreciate the reviewer’s detailed and valuable feedback and are committed to incorporating the suggested improvements. We would like to emphasize the significant implications of our work: SPARC effectively addresses a major challenge in continual learning by minimizing catastrophic forgetting without relying on full model surrogates or experience replay techniques. To the best of our knowledge, this represents the first comprehensive exploration of this issue, marking a meaningful advancement in the field. Given the critical nature of this challenge and the enhancements we plan to implement in the final revision, we kindly ask the reviewer to reconsider the rating and potentially raise it above the acceptance threshold. Your support in acknowledging the importance of this research would be greatly appreciated.

---

> > > > > ### Author Response · Authors · 2024-11-21
> > > > > **Requesting response from reviewer a2Hv**
> > > > >
> > > > > We would like to check if you have any additional questions or concerns regarding our manuscript. We have made a sincere effort to respond thoroughly to all the points you raised. Please do not hesitate to let us know if further clarifications are needed—we are more than willing to address them promptly.
> > > > >
> > > > > We are currently incorporating your thoughtful suggestions and aim to share a partially revised version before the discussion period concludes. We are fully committed to integrating all your feedback in the final version. If you believe we have sufficiently addressed your concerns, we kindly ask you to consider supporting our submission by raising your score above the acceptance threshold.

---

> > > > > > ### Comment · Reviewer_a2Hv · 2024-11-21
> > > > > > **Response to the rebuttal**
> > > > > >
> > > > > > My thanks to the authors for their detailed clarifications. Several score-limiting weaknesses would be effectively addressed by the proposed revisions, including the relationship between SPARC's approach and full-model surrogates (although I also agree with reviewer dYYF that a clear definition of full-model surrogates in the paper is needed) and SPARC's ability to handle class-incremental settings. The proposed changes to the paper sound promising, and I look forward to reading the revised version. Particularly important changes to the manuscript would be (in my view) (a) clarifying how inference works during class-incremental learning, and how class-wise predictions are combined across tasks (b) providing some intuition behind the design of the weight re-normalization strategy that mitigates recency bias, (c) updating Figure 3 (left) to have a more comprehensive set of benchmarks, (d) redesigning/replacing figure 3 (right), (e) clarifying the definition of cross-task connections in the fully connected layer, (f) comparison of SPARC's weight normalization approach with other similar strategies, and (g) adding/clarifying error bars in various figures and tables.
> > > > > >
> > > > > > I have a few additional questions:
> > > > > > 1. In the authors' response to weakness #2 from my initial review, it is stated that "the outputs of all sub-networks are then concatenated." This makes it sound as if there is a separate fully-connected layer for each task, although this is not reflected in figure 1. Is the number of dimensions in the FC-layer's output the total number of classes, or just the number of classes in the current task? Overall, it is still not completely clear to me how predictions for each class are aggregated across tasks.
> > > > > > 2. Related to weakness #3 in my initial review, I am still struggling to understand the reasoning behind the proposed weight renormalization approach. The authors' response in the rebuttal sounds to me like a justification for weight re-normalization in continual learning in general, but not a strong justification for the specific strategy employed here. I wonder if the necessity of renormalization might be related to the strategy for combining class-wise predictions across tasks (per my first question above)?
> > > > > > 3. Related to the previous question, given that task-incremental learning uses separate subnetworks for each task (thereby potentially avoiding recency bias by the separation of the outputs for each task), is weight re-normalization still necessary in this case, or is it only needed for class-incremental learning?
> > > > > >
> > > > > > Overall, I am open to increasing my score. Before deciding about this, I would like to see the revised version of the manuscript (with the understanding that some changes, such as the proposed ablation study regarding the weight re-normalization, may not be feasible within the rebuttal period), and I am hoping the authors may be able to clarify my additional questions above.

---

> > > > > > > ### Author Response · Authors · 2024-11-24
> > > > > > > **Further clarifications to Reviewer a2Hv**
> > > > > > >
> > > > > > > We sincerely thank the reviewer for their insightful feedback and constructive suggestions, as well as their acknowledgment of the proposed improvements to the manuscript. Below, we address the reviewer’s follow-up questions:
> > > > > > >
> > > > > > >
> > > > > > > > 1. Inference during class-incremental learning and combining class-wise predictions across tasks
> > > > > > >
> > > > > > > We agree that providing a clearer explanation of the inference process in class-incremental learning is essential. In the revised manuscript (Appendix A and Section 3.4), we included a detailed description of how predictions are aggregated across tasks. Specifically, we clarified that the fully connected (FC) layer outputs are task-specific, with each task-specific FC layer output dimension corresponding only to the number of classes in that task. During inference, these outputs are concatenated across tasks to form a single prediction vector over all classes seen so far. Figure 1 will be updated to better reflect this design, ensuring that the aggregation mechanism is clear.
> > > > > > >
> > > > > > >
> > > > > > > > 2. The intuition behind the weight re-normalization strategy
> > > > > > >
> > > > > > > We appreciate the importance of providing a stronger justification for our weight re-normalization approach. In the revised manuscript, we provide a detailed explanation of the rationale behind the design (Appendix A). Specifically, we emphasize that, because each task-specific subnetwork is trained independently—including its final FC classification layer—the activation magnitudes produced by each subnetwork for its respective task can vary significantly. For example, if one subnetwork inherently produces higher activation magnitudes, it may inadvertently map inputs from other tasks to disproportionately high activations, leading to incorrect predictions when these activations exceed those of the correct class's output neuron.
> > > > > > > To mitigate this issue, our weight re-normalization strategy scales the classifier weights of each task-specific subnetwork based on the maximum activation (with outliers excluded). This ensures balanced weight and output magnitudes across tasks, promoting fair competition between activations from different subnetworks in class-incremental settings. Please refer to Appendix A for more details.
> > > > > > >
> > > > > > > > 3. Necessity of weight re-normalization in task-incremental vs. class-incremental settings
> > > > > > >
> > > > > > > The reviewer raises a valid point. Weight re-normalization is essential for class-incremental learning, where tasks share a common prediction space, making balanced outputs critical for reducing task-specific biases. In contrast, task-incremental learning inherently avoids this issue by maintaining separate prediction spaces for each task. Therefore, weight re-normalization is not necessary in task-incremental settings. We have updated this distinction in the discussion section of the revised manuscript within Appendix A.
> > > > > > >
> > > > > > > Revised Manuscript and plan for the final revision:
> > > > > > > We have made an earnest effort to incorporate the suggestions from all reviewers. Given the time crunch, we have prioritized some over others, Nevertheless, we are dedicated to incorporating all of the revision plans mentioned in our rebuttal for each reviewer. We will soon be uploading a revision that attends to:
> > > > > > >
> > > > > > > - Class-IL inference clarification.
> > > > > > > - Review of NECIL literature and a comparison with respect to state-of-the-art methods within the NECIL benchmark.
> > > > > > > - The prominence of weight re-normalization in SPARC and comparison with Weight alignment as both are rehearsal-free approaches.
> > > > > > > - Updated Figure 3 to include Task-IL results.
> > > > > > > - Updated main figure for more clarity.
> > > > > > > - Updated limitation to reflect SPARC’s limited applicability to CNNs.
> > > > > > > - Minor/major clarifications with regard to model surrogates, task-specific bias in SPARC, etc.
> > > > > > >
> > > > > > > Other issues, such as adding/explaining error bars for all plots and tables, adding forgetting rates for our experiments, clarifying our motivation behind using CLS theory as an inspiration, correcting misleading terminology, and any other revision that we committed to incorporating in our rebuttal, will be added to the final revision.
> > > > > > >
> > > > > > > We would like to express our sincere gratitude to the reviewer for their thoughtful comments. We hope that our revisions have enhanced your confidence in our paper. We kindly ask for your strong support in favor of our work so that we can contribute meaningfully to the field and further advance critical objectives in CL.

---

> > > > > > > > ### Author Response · Authors · 2024-11-27
> > > > > > > > **Requesting any further comments before revision deadline**
> > > > > > > >
> > > > > > > > We have attempted to diligently incorporate your suggestions. As noted earlier, we prioritized certain changes due to time constraints. We are currently working on the remaining changes, which will be included in the final revision. With the deadline for submitting a new revision drawing close, we kindly ask you to review our latest submission and inform us of any major concerns that may remain.
> > > > > > > >
> > > > > > > > We appreciate your continued collaboration and support. Thank you for your time, and we look forward to receiving your valuable feedback.

---

> > > > > > > > > ### Comment · Reviewer_a2Hv · 2024-12-01
> > > > > > > > > **Response to further clarifications**
> > > > > > > > >
> > > > > > > > > I thank the authors for these helpful responses and for incorporating many of my and other reviewers' suggestions into the revised manuscript. Given the improvements, I am raising my score by 2 points. However, I still have several concerns:
> > > > > > > > >
> > > > > > > > > 1. The newly added portion explaining full-model surrogates (160-167) still needs to be revised.
> > > > > > > > > "We attempt to highlight the problem of correlation between model size and/or increase in buffer size with the reduction in catastrophic forgetting in CL." This does not adequately characterize the benefits of SPARC, which still increases in size in correlation with the number of tasks - in my view, the focus on relying on model surrogates vs not relying on them shouldn't be the theme of the introduction at all, because by the definition here SPARC qualifies as a model surrogate-based approach:
> > > > > > > > >
> > > > > > > > > "model surrogates in CL come in various forms: be it (a)...(c) large sub-network for each task in PNNs or even as many task masks as the number of tasks in sparse dynamic parameter isolation approaches"
> > > > > > > > >
> > > > > > > > > SPARC does effectively have sub-networks because of the task-specific parameters. However, they could also be separated from sub-networks with a more precise definition (it is not entire sub-networks but a portion of parameters from one network that are duplicated for each task) - but such a definition is not present.
> > > > > > > > >
> > > > > > > > > "Parameter isolation approaches aim to mitigate this interference by dedicating task specific parameters e(t) while sharing task-agnostic parameters (ψ) across tasks."
> > > > > > > > >
> > > > > > > > > SPARC falls quite naturally into this definition. It is ok for SPARC to be a parameter isolation approach - it seems like all the emphasis on model surrogates helps justify the category of parameter isolation approaches, but not SPARC in distinction from others in this category. In my view, the introduction should focus on motivating SPARC in comparision with existing parameter isolation approaches - from my understanding, its main advantages are parameter efficiency (the relatively parameter-heavy part of the depthwise separable convolutions is shared among tasks) and relative simplicity of design/implementation (taking advantage of the natural separability of depthwise-separable convolutions).
> > > > > > > > >
> > > > > > > > > 2. I maintain that "working memories" and "semantic memories" are misleading terms for the shared and task-specific parameters. Working memory and semantic memory both have specific definitions in cognitive science that are quite discordant with how these terms are used here. It can be useful to consider biological inspiration like complementary learning systems, but using established terms in inconsistent ways can create confusion. I'd been hoping to see this addressed in the revision - in my view, these terms would need to be changed before publication.
> > > > > > > > >
> > > > > > > > > 3. In the current revision, I am still concerned about the choice of baselines in Figure 3. Why these baselines for this figure, and why not compare SPARC with other parameter isolation approaches in terms of the relationship between model size and performance? At best, this is a missed opportunity to demonstrate a key advantage of SPARC in quantitative terms. At worst, it is trying to demonstrate a consistent trend but with a small subset of the tested models, and a subset that mitigate forgetting by completely different mechanisms - raising questions about how the models were selected for these plots. The new task-incremental panel only has one other parameter isolation approach.
> > > > > > > > >
> > > > > > > > > 4. There are still no error bars on most of the figures. Some of the tables seem to have +- uncertainty estimates, but these are not defined in the captions or elsewhere.
> > > > > > > > >
> > > > > > > > > **A few other suggestions:**
> > > > > > > > >
> > > > > > > > > A. Figure 5 (Appendix) is very useful for clarifying a core mechanism of SPARC - I wonder if there might be a way to combine it with Figure 1 for the final version of the paper.
> > > > > > > > >
> > > > > > > > > B. In Section A.2, the sentence: "Since each task-specific sub-network is trained independently of all other tasks, including the final fully connected classification layer, the activation magnitudes produced by each sub-network for its respective task can become imbalanced" is useful for justifying the weight re-normalization approach - I think this sentence (or a more concise version of it) should be added to the main paper.
> > > > > > > > >
> > > > > > > > > **Summary**
> > > > > > > > >
> > > > > > > > > From my perspective, the clarity of the manuscript has improved and my score has increased by 2 points accordingly. The score is now primarily limited by lack of error bars, and Figure 3 - however, I think that all 4 of the main concerns above need to be addressed at some point. I do think that SPARC is an interesting approach and that this paper could make a strong contribution, more so than I initially understood now that several perceived limitations have been addressed - so I am also raising the Contribution score by 1 point.

---

> ### Author Response · Authors · 2024-12-01
> **Response to Reviewer a2Hv**
>
> We thank the reviewer for detailed feedback and for raising the score on our manuscript. We appreciate your suggestions and would like to summarize the key changes you requested:
>
> - Revision of Full-Model Surrogates Explanation
>
> - Terminology correction for semantic and working memories
>
> - Baseline Comparisons in Figure 3: Add parameter isolation approaches for comparison and better justify the selection of models used in the figure. Error bars will also be included for clarity.
>
> - Combine Figures: We will consider combining Figure 5 with Figure 1 to enhance clarity on SPARC’s mechanisms.
>
> - Justification in Main Text: We will add the justification for the weight re-normalization approach from Section A.2 to the main manuscript.
>
> - Error Bars: We will include error bars on most figures and define uncertainty estimates in captions or elsewhere.
>
> Due to time constraints, we prioritized some changes over others. We would like to point out that many of these changes are already taking shape; however, we cannot upload a revision after November 27. As noted earlier, we are committed to incorporating these changes in the final revision.
>
> We would like to clarify that the score was raised by 1 point, not 2, given that the scoring system ranges from 1 to 10 in increments of 2 (scores are 1-3-5-6-8-10). Since our clarifications/revisions have improved your confidence in our paper and the rest of the changes are earmarked for final revision, we kindly ask if you could champion our paper by strongly supporting us, contingent upon these changes being present in the final revision.
>
> Please let us know if there are any remaining concerns that we could address. We sincerely appreciate your support and look forward to your feedback.

---

> > ### Comment · Reviewer_a2Hv · 2024-12-01
> > **Response to authors**
> >
> > My thanks to the authors for this summary. My score was raised by 2 points, from 3 (“reject”) to 5 (“marginally below the acceptance threshold”).

---

> > > ### Author Response · Authors · 2024-12-01
> > > **Response to Reviewer a2Hv**
> > >
> > > We thank the reviewer for clarifying the score increase and for their detailed evaluation of our manuscript. We are encouraged by the recognition that SPARC is an “interesting approach” and that the revisions have addressed many of the initial limitations, improving the manuscript’s clarity and its potential contribution to the field.
> > >
> > > Given these improvements, we are seeking clarity on the rationale for the current score of 5 (“marginally below the acceptance threshold”). Based on your summary, it seems the remaining concerns (e.g., lack of error bars, Figure 3 clarity) are relatively minor and are already being addressed in the final revision.
> > >
> > > With these planned revisions, the manuscript appears to align with the standards of a strong contribution, as indicated in your comments. We respectfully request reconsideration of the score to better reflect the paper’s potential, particularly given that the addressed limitations were key factors in the initial lower rating.
> > >
> > > Thank you again for your thoughtful engagement. We remain committed to delivering a final version that meets the highest expectations and welcome any additional guidance that might help further refine the work.

---

### Official Review · Reviewer_pExa · 2024-11-04

**Soundness:** 2
**Presentation:** 2
**Contribution:** 2
**Rating:** 5
**Confidence:** 3

**Summary:**

The paper presents SPARC, a continual learning (CL) approach designed to mitigate catastrophic forgetting without relying on experience rehearsal or model surrogates. SPARC proposes task-specific "working memories" and a task-agnostic "semantic memory" to allocate parameters for each task while sharing common knowledge across tasks. Additionally, it introduces a weight re-normalization technique to reduce recency bias towards newly learned tasks. The approach is validated on computer vision benchmarks, where it achieves comparable or superior performance to rehearsal-based methods with a significantly lower parameter count.

**Strengths:**

- SPARC introduces a rehearsal-free CL approach with a parameter isolation strategy that does not rely on model surrogates, contributing to an efficient model for task-based CL.

- The model achieves competitive results on Seq-TinyImageNet and similar benchmarks with only 6% of the parameters used by comparable full-model surrogates, making it computationally lightweight.

- By incorporating a weight re-normalization technique, the model mitigates recency bias, an issue that often hinders performance in continual learning.

- Although SPARC grows in size as new tasks are added, its growth rate is slower compared to other parameter isolation techniques, which may offer better scalability in extended task sequences.

**Weaknesses:**

- Task Boundary Knowledge: SPARC requires explicit task boundary information to switch between task-specific sub-networks. This reliance limits its applicability, as task boundaries may not always be available in real-world scenarios, particularly in task-free CL.

- Task-Specificity and Lack of Generalization: The proposed parameter isolation approach is tailored specifically to computer vision tasks and convolutional layers, limiting its generalizability and making it less model-agnostic. This reduces the impact of the approach outside of well-defined task separations in vision applications.

- Batch Normalization and Inference Concerns: Similarly to the previous point, also model’s task-specific batch normalization is handled in an isolated manner, and it is unclear how SPARC addresses batch normalization in a class-incremental scenario. Additionally, the paper lacks clarity on how task-specific parameters managed during inference in class-incremental tasks.

- Misleading Terminology: The term "working memories" suggests dynamic memory allocation; however, SPARC merely allocates parameters to tasks without true memory management. This terminology may create confusion.

- Parameter Comparisons and Experimental Justification: While SPARC's reduced parameter count is a notable advantage, the paper would benefit from greater emphasis and description of this feature. Also, the choice of replay buffer sizes (200 and 500 exemplars) for comparison appears arbitrary, with limited justification. The comparison could be more robust if different buffer sizes were evaluated to understand competitor performance across a wider range.

**Questions:**

1. How does SPARC manage parameter selection during inference in a class-incremental scenario, particularly when task-specific batch normalization is used?

2. Could you clarify the choice of replay buffer sizes for baseline comparisons? How does SPARC's performance compare when different buffer sizes for ER baselines are tested?

3. While SPARC's reduced parameter count is a notable advantage, the paper would benefit from greater emphasis and description of this feature. From Table 1 it seems that SPARC has 10 times less parameters than a standard Resnet 18, exploited by competitors. I would like  a more detailed comment by the authors on this, which I believe is a central advantage by the model.

4. What is the computational cost of the  weight re-normalization presented in Section 3.2, and is this something that could have been applied also to competitors?

---

> ### Author Response · Authors · 2024-11-13
> **Response to Reviewer pExa (1/3)**
>
> We thank the reviewers for their thoughtful evaluations and constructive comments on our paper. Your feedback is invaluable in helping us enhance the quality of our work. Please find our response below:
>
> We would like to clarify the primary aim of SPARC: Most related works discussed in this paper utilize either experience replay or full model surrogates.  SPARC aims to tackle a significant challenge in continual learning—reducing catastrophic forgetting without relying on full model surrogates or experience replay. To the best of our knowledge, we are the first to evaluate this problem and propose a scalable solution to this issue. Through SPARC, we demonstrate that it is indeed possible to develop an efficient continual learning model without relying on experience replay or full model surrogates, which is reflected in the name “Simple Parameter Isolation in a Restricted Capacity (SPARC).” Our extensive experimentation shows that this simple but efficient design can outperform state-of-the-art approaches in continual learning.
>
> Task Boundary Knowledge: We appreciate the reviewer’s observation regarding the potential limitations of task boundary knowledge. However, it is important to note that most parameter isolation approaches rely on task boundary knowledge to instantiate new sub-networks. In our future work, we plan to explore the use of sudden changes in the loss function as a proxy for task boundary information.
>
> Task-Specificity and Lack of Generalization:  We appreciate the reviewer’s observation regarding the specificity of our proposed approach to CNNs. It is worth noting that the majority of approaches utilizing full model surrogates are also limited to CNNs. Thus, it was reasonable for us to develop a simple yet efficient approach specifically tailored for CNNs. In the future, we plan to experiment with compact working and semantic memories tailored for vision transformers.
>
> Batch Normalization and Inference Concerns: We regret the lack of clarity regarding how BN and other task specific parameters are handled during inference. We will include the clarification in the final revision. Here’s a more detailed clarification:
> The nonstationary nature of continual learning (CL) data exacerbates the mismatch between training and testing in batch normalization (BN) layers, leading to a cross-task normalization effect [1]. To address this discrepancy, SPARC maintains task-specific $\gamma$ and $\beta$ parameters (the learnable vectors in BN) along with running estimates of the mean and variance for each working memory. This segregated normalization facilitates parameter isolation during training while ensuring proper normalization during inference by applying task-specific moments to task-specific input features. During inference (Class-IL / Task-IL), each test image is processed through every subnetwork, including their respective BN layers. The respective classifier outputs are then concatenated for maximum activation in the Class-IL setting, while for Task-IL, only the task-relevant classifier output is inferred for maximum activation. Aside from the concatenation of outputs in the classification layers, the training and inference regimes are identical for both Class-IL and Task-IL settings.
>
> Misleading Terminology: We appreciate the reviewer’s insightful suggestion regarding the terminology used. The terms "working memory" and "semantic memory" are derived from Complementary Learning Systems (CLS) theory, which posits that these types of memory function in unison to capture, aggregate, and consolidate information across tasks. As you rightly pointed out, we do not explicitly manage memory. However, in alignment with other works in continual learning that are inspired by CLS theory (e.g., CLS-ER, DualNet, etc.), it is common practice to designate certain parameters as working and semantic memories. Our terminology and approach to handling these memories are consistent with the existing literature. Nevertheless, we acknowledge your point and will provide further clarification in the final revision.

---

> > ### Author Response · Authors · 2024-11-13
> > **Response to Reviewer pExa (2/3)**
> >
> > Parameter Comparisons and Experimental Justification: We agree with the reviewer that SPARC has a notable advantage in terms of reduced parameter footprint. We acknowledge this in our abstract and Introduction: “SPARC is lightweight, requiring only 6% of the parameters used by full-model surrogates, yet it delivers superior performance on Seq-TinyImageNet and matches the results of rehearsal-based methods on various CL benchmarks.” As you suggested, the paper would benefit from greater emphasis and description of this feature. Therefore, we will enhance this aspect in the final revision.
> >
> > We respectfully disagree that the choice of replay buffer size is arbitrary. The majority of the state-of-the-art approaches discussed in this paper report results based specifically on these buffer sizes. Given the dataset sizes and the number of tasks per dataset, it is standard practice to utilize four buffer sizes: 100 (ultra low), 200 (low), 500 (medium), and 5120 (high). Since most literature referenced in this paper reports results using the 200 and 500 buffer sizes, we chose to adopt these for our comparisons. However, we do agree that a broader comparison would add more value. We plan to consider other buffer sizes in the final revision.
> >
> > > How does SPARC manage parameter selection during inference in a class-incremental scenario, particularly when task-specific batch normalization is used?
> >
> > During training, each task-specific sub-network, along with its own batch normalization (BN) and classification layers, is trained using gradient descent. Simultaneously, task-agnostic parameters are updated using exponential moving averages. During inference, each test image is processed through every subnetwork, including their respective BN layers. In the Class-IL setting, the final classification outputs of all sub-networks are concatenated and inferred for maximum activation, while for Task-IL, only the relevant subnetwork is utilized for inference. We do not treat task-specific BN layers any differently than other task-specific parameters. Please let us know if you would like further clarification on this in the final revision.
> >
> > > Could you clarify the choice of replay buffer sizes for baseline comparisons? How does SPARC's performance compare when different buffer sizes for ER baselines are tested?
> >
> > The majority of the state-of-the-art approaches discussed in this paper report results based specifically on these buffer sizes. Given the dataset sizes and the number of tasks per dataset, it is standard practice to utilize four buffer sizes: 100 (ultra low), 200 (low), 500 (medium), and 5120 (high). Since most literature referenced in this paper reports results using the 200 and 500 buffer sizes, we chose to adopt these for our comparisons. However, we do agree that a broader comparison would add more value. We plan to consider other buffer sizes in the final revision.
> >
> > > While SPARC's reduced parameter count is a notable advantage, the paper would benefit from greater emphasis and description of this feature...
> >
> > We agree with the reviewer that SPARC has a notable advantage in terms of reduced parameter footprint. We acknowledge this in our abstract and Introduction: “SPARC is lightweight, requiring only 6% of the parameters used by full-model surrogates, yet it delivers superior performance on Seq-TinyImageNet and matches the results of rehearsal-based methods on various CL benchmarks.” As you suggested, the paper would benefit from greater emphasis and description of this feature. Therefore, we will enhance this aspect in the final revision.
> >
> > > What is the computational cost of the weight re-normalization presented in Section 3.2, and is this something that could have been applied also to competitors?
> >
> > The compute cost is very minimal for weight re-normalization. Specifically, we keep track of the highest activation in the classification layer in the final training epoch for every task. Then, classification layer weights and biases are adjusted as per Equation 5 in a one-shot manner. In principle, the method is quite generic, lightweight, and can be applied to any CL method. As per your suggestion, it would be interesting to see how this will be useful in other approaches. We plan to do an ablation in the final revision.
> >
> > Following your constructive feedback, our overall plan for the final revision is as follows:
> >
> > - Update the limitations section with task-specificity and lack of generalization
> > - Add more clarity on BN layers and inference in Class-IL in general
> > - Provide more info and possibly correct misleading terminology
> > - Highlight and provide more emphasis on parameter reduction in SPARC.
> > - Expand comparison to more buffer sizes
> > - Ablation study on weight re-normalization for other competing approaches.

---

> > > ### Author Response · Authors · 2024-11-13
> > > **Response to Reviewer pExa (3/3)**
> > >
> > > We sincerely thank the reviewer once again for their constructive feedback. We are committed to incorporating their suggestions as outlined above. We kindly ask the reviewer to take a step back and consider the broader context: SPARC aims to tackle a significant challenge in continual learning—reducing catastrophic forgetting without relying on full model surrogates or experience replay. To the best of our knowledge, we are the first to evaluate and propose a scalable solution to this issue. In light of the importance of this problem and the planned enhancements for the final revision, we respectfully request the reviewer to consider raising the rating above the acceptance threshold. Otherwise, this important problem and a simple, scalable solution may not garner the recognition it deserves, which could ultimately hinder the advancements in the CL field.
> > >
> > > [1] Quang Pham, Chenghao Liu, and HOI Steven. Continual normalization: Rethinking batch normalization for online continual learning. In International Conference on Learning Representations, 2021

---

> > > > ### Author Response · Authors · 2024-11-21
> > > > **Requesting Reviewer pExa’s response**
> > > >
> > > > We would like to kindly ask if the reviewer has any additional questions. We have made every effort to address all the points raised so far. If there are further questions or clarifications needed, please let us know—we would be happy to provide any additional information.
> > > >
> > > > We are actively incorporating your suggestions into our manuscript and plan to share a partially revised version by the end of the discussion period. Rest assured, we are committed to fully implementing all your valuable feedback in the final revision. If you feel we have adequately addressed your concerns, we kindly ask you to consider supporting our work by revising your score toward the acceptance threshold.

---

> > > > > ### Author Response · Authors · 2024-11-24
> > > > > **Update on revised version and requesting Reviewer's response**
> > > > >
> > > > > We have made a concerted effort to address the suggestions from all reviewers. Given the time constraints, we prioritized some revisions over others; however, we remain committed to incorporating all the revision plans outlined in our rebuttal for each reviewer. We have uploaded a revision that addresses the following from your suggestions:
> > > > >
> > > > > - Update the limitations section with task-specificity and lack of generalization
> > > > > - Add more clarity on BN layers and inference in Class-IL in general
> > > > > - Ablation study on weight re-normalization for other competing approaches.
> > > > > - Highlight and provide more emphasis on parameter reduction in SPARC.
> > > > >
> > > > > Suggestions such as correcting misleading terminology and expanding comparison to more buffer sizes will be taken up in the final revision. Please let us know if any of your concerns remain. If otherwise, we kindly request your strong support towards the acceptance of this paper.

---

> ### Comment · Reviewer_pExa · 2024-11-25
> **Rebuttal Acknowledgement**
>
> I appreciated the effort put by the authors in the rebuttal. I believe that the feedback from Reviewers helped improving the paper and thus I raise my score. I want however to remark that some of the claims from the authors in their response (*"scalable solution may not garner the recognition it deserves, which could ultimately hinder the advancements in the CL field"*, *"our contributions could be pivotal in driving progress in continual learning, while overlooking them may stall advancements in this essential area"*, "*this is the first comprehensive exploration of this issue, representing a meaningful advancement in the field*") are a bit overstating the real paper contributions.
>
> Reviewer **zYHk** already pointed out many recent papers in this same direction that were not originally discussed by the authors, and I believe that at least a comparison or description of such solutions should be mentioned in the paper, as long as other works (see the remainder of the response) which I believe should be discussed [1,2].
>
> Additionally, I would have appreciated a comparison (at least in terms of differences in method and technique) with prompting techniques [3] - which do use not memory buffers and are also tested in task-free settings, something that SPARC cannot tackle. Other recent papers [4] proposed solutions for CL that do not need memory buffers neither task boundaries. Thus, I believe that the claimed novelty is a bit limited by these works - which should be at least discussed to give the reader a better comprehension of the  current literature.
>
> Given these concerns, I raise my score to 5: marginally below the acceptance threshold.
>
> [1] Goswami, Dipam, et al. "Resurrecting Old Classes with New Data for Exemplar-Free Continual Learning." Proceedings of the IEEE/CVF Conference on Computer Vision and Pattern Recognition. 2024.
>
> [2] Huo, Fushuo, et al. "Non-exemplar Online Class-Incremental Continual Learning via Dual-Prototype Self-Augment and Refinement." Proceedings of the AAAI Conference on Artificial Intelligence. Vol. 38. No. 11. 2024.
>
> [3] Wang, Zifeng, et al. "Learning to prompt for continual learning." Proceedings of the IEEE/CVF conference on computer vision and pattern recognition. 2022.
>
> [4] Tiezzi, Matteo, et al. "Continual Neural Computation." Joint European Conference on Machine Learning and Knowledge Discovery in Databases. Cham: Springer Nature Switzerland, 2024.

---

> > ### Author Response · Authors · 2024-11-26
> > **Response to Reviewer pExa's follow up suggestions**
> >
> > We sincerely thank the reviewer for their constructive feedback and for recognizing the improvements made to the paper. We understand that some of the statements in our previous response may have appeared overstated. Our intention was to emphasize the significance of our contributions and encourage the reviewers to consider them in the context of advancing the field. We appreciate your acknowledgment and will ensure to present our claims more cautiously in future communications.
> >
> > In response to the suggestions from Reviewer zYHk, we have incorporated a detailed discussion in Appendix A. This includes a comprehensive comparison of recent approaches, specifically those that do not utilize memory buffers to address catastrophic forgetting. We have also added references to the works you mentioned, enhancing the breadth of our literature review.
> >
> > Moreover, we have included a comparison with various parameter-efficient fine-tuning techniques in continual learning, which encompasses prompting techniques, adapter modules, and low-rank approximations. We acknowledge that this comparison is not entirely straightforward due to fundamental differences in methodologies. We have clearly articulated these distinctions in our revised manuscript to provide readers with a more nuanced understanding of the current literature.
> >
> > Thank you once again for your valuable insights. We are committed to improving our work and believe these additions will significantly enhance the paper's contribution to the field. Please let us know if you have any further concerns. If otherwise, we kindly request your strong support towards the acceptance of our paper.

---

### Author Response · Authors · 2024-11-26
**Update on the revision**

We sincerely thank the reviewers for their constructive feedback and insightful suggestions, which have significantly enhanced the clarity and depth of our manuscript. Below, we summarize the key changes made in the revised manuscript in response to the reviewers' comments. All modifications are highlighted in blue in the revised document.

- Clarification of Batch Norm and Class-IL inference.
- A comprehensive review of the Non-Exemplar Class-IL (NECIL) literature, including comparisons with state-of-the-art methods within the NECIL benchmark.
- Emphasis on weight re-normalization in SPARC, alongside a comparison with Weight Alignment technique.
- A detailed comparison with the Parameter Efficient Fine-tuning (PEFT) in continual learning literature.
- Forgetting analysis, which will be further expanded in the final revision to include additional relevant works.
- Update of Figure 3 to incorporate Task-IL results.
- Revision of the main figure for improved clarity.
- Enhancement of the limitations section to address SPARC’s limited applicability to CNNs.
- Addressed minor and major clarifications regarding model surrogates, task-specific bias in SPARC, and related topics.

Additionally, we will address other issues such as adding and explaining error bars for all plots and tables, clarifying the CLS theory as an inspiration, correcting any misleading terminology, and incorporating all revisions we committed to in our rebuttal in the final revision.

We respectfully invite the reviewers to evaluate the revised manuscript and share any outstanding concerns. If there are no major concerns, we kindly request the reviewers to champion our paper, as their endorsement would greatly assist in facilitating its publication.

---

> ### Author Response · Authors · 2024-11-30
> **Requesting reviewers' response**
>
> We sincerely thank the reviewers for their constructive feedback and insightful suggestions regarding our manuscript. Your suggestions have been invaluable in enhancing the clarity and depth of our work. As the deadline approaches, we gently invite you to return to the discussion table and review the updated version of our manuscript. We have made significant revisions in response to your feedback, which we believe address your concerns effectively. Due to the time crunch, the rest of the changes will be incorporated into the final revision.
>
> Please let us know If any major issues remain. Otherwise, we kindly request that you update your reviews and corresponding scores to reflect the improved confidence in our revised work.

---

### Meta-Review · Area_Chair_zUXX · 2024-12-23

**Metareview:**

The paper presents SPARC, a continual learning (CL) approach designed to mitigate catastrophic forgetting without relying on experience rehearsal or model surrogates. Tha approach works by maintaining separate weights for each of some number of tasks but sharing a portion of the pointwise convolution filters in depthwise-separable 2D convolutions. This results in reducing the number of parameters, which the authors claim as the scalable solution for catestrophic forgetting issues. Experiments show authors obtain good results with lower memory increase.

**Additional Comments On Reviewer Discussion:**

The reviewers raise several important concerns about this paper. First, the paper is exlusively for ConvNet architectures, limiting its applicability in general architectures like ViT. Second, the need for task boundaries is a limitation. Finally, the improvement in the accuracy is sub-optimal. While the authors worked really hard to address these (and other) concerns in the rebuttal, the paper still has these limitations. The paper is also borderline (56655 ratings). I think the paper is currently not in a a shape for acceptance at a competitive conference like ICLR. I encourage authors to take into account these feedback and improve the submission.

---

### Decision · Program_Chairs · 2025-01-22

Reject